# BEYOND SCORE: A MULTI-AGENT SYSTEM TO DISCOVER CAPABILITY AND BEHAVIORAL WEAKNESSES IN LLMS

## ABSTRACT

A key task for researchers working on large language models (LLMs) is to compare the results and behavioral performance of different models, thereby identifying model weaknesses and enabling further model improvements. However, as LLMs are applied in an increasing range of scenarios and the number of benchmarks continues to grow, the difficulty of accurately identifying weaknesses increases. Additionally, with the emergence of Reasoning LLMs, researchers need to analyze the chain-of-thought (CoT) behaviors of models to gain insights—this makes the task of directly analyzing model capabilities based on benchmark evaluation results more onerous and unreliable. To address these issues, we propose AGENT4WEAKNESS, a framework that uses multi-agent collaboration to generate evaluation reports with user requirements for LLM evaluation. Specifically, AGENT4WEAKNESS employs multiple mainstream LLMs for evaluation and comparison, incorporating professional statistical tools to provide richer statistical insights. Besides, AGENT4WEAKNESS features a dedicated agent designed to extract relevant information from the results according to user requirements, ensuring the final analysis is tailored to user needs. We show that reports generated by AGENT4WEAKNESS achieve an improvement of 2.6 out of 10 across four dimensions compared with the baseline, with high consistency with human evaluations, which proves the high quality of the reports. Furthermore, guided by the reports from AGENT4WEAKNESS, we achieve a 3.7 performance gain by addressing the discovered weaknesses via targeted prompt-level interventions, demonstrating the significant practical value of AGENT4WEAKNESS.

## 1 INTRODUCTION

As large language models (LLMs) advance, the evaluation to identify model weaknesses becomes increasingly crucial, which we call **Weakness Discovery** (Zhao et al., 2024; Zeng et al., 2025). This process aids in the understanding, development, and improvement of current LLMs (Chang et al., 2024; Peng et al., 2024). Specifically, weakness discovery conducts a deeper analysis of direct evaluation data (e.g., response, accuracy) to identify more specific differences in capabilities on certain models or datasets (Chang & Bergen, 2024; Hu & Zhou, 2024). Previous weakness discovery work can be categorized into two types. One approach involves encoding and clustering questions or model outputs, identifying clusters with low performance as weaknesses (Zeng et al., 2025; Tian et al., 2025; Lee et al., 2025). The other approach analyzes weaknesses of the model on an instance-by-instance basis (Murahari et al., 2024; Yang et al., 2024; Moayeri et al., 2025).

However, as shown in Figure 1, existing weakness discovery methods exhibit two primary limitations: *(i) Insufficient Comparison*: Current methods primarily report performance differences without analyzing the nature of these disparities, such as their statistical significance and confidence, which limits the substantive value of the evaluation results (Mizrahi et al., 2024; Luettgau et al., 2025). *(ii) Inflexible Evaluation*: Current works are restricted to fixed evaluation perspectives and lack the flexibility to generate diverse results based on user requirements, which limits the general applicability of such methods (Brawer et al., 2023).

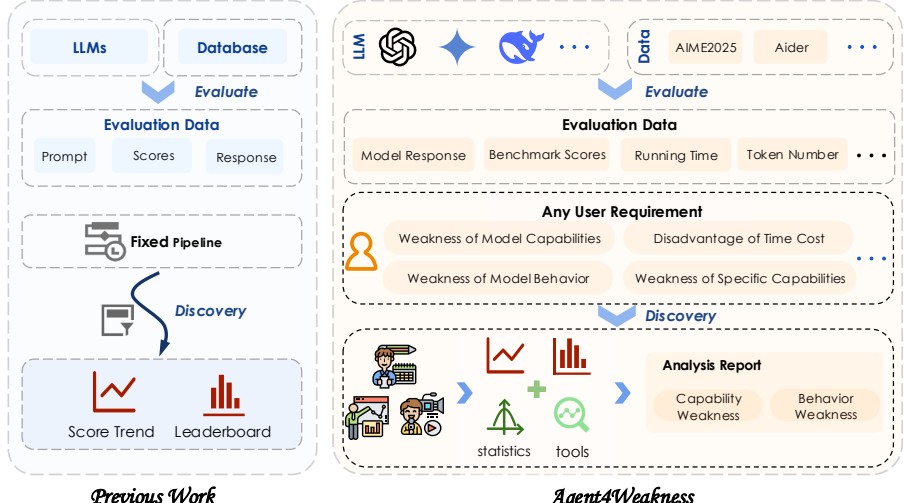

Figure 1: The comparison between the previous weakness discovery method (left) and our work (right). Previous works lack statistical analysis of evaluation data and are limited to generating analyses from fixed perspectives. In contrast, our work provides richer statistical analyses and aligns its evaluation with user requirements, thereby demonstrating higher reliability and flexibility.

Given these shortcomings, we argue that an effective weakness discovery method should satisfy the following criteria: *(i) Sufficient Comparison*: It should not only identify performance differences but also assess their nature, like significance and confidence, to ensure the findings are substantive. *(ii) Flexible Evaluation*: It should be capable of generating customized evaluation results for specific aspects according to user needs, ensuring high generalizability.

Based on the above analysis, we propose AGENT4WEAKNESS, a multi-agent system equipped with diverse customized tools (Table 1) that perform weakness discovery based on user queries. The illustration of AGENT4WEAKNESS is shown in Figure 1. To provide substantive value, we incorporate professional statistical tools, making the discovered weaknesses more general and robust. AGENT4WEAKNESS also analyzes user needs and retrieves relevant information to generate customized evaluation results, flexibly meeting user requirements.

To validate the effectiveness of AGENT4WEAKNESS, we first evaluate 104 models on 27 datasets, from which we select 8 representative models for analyzing weakness using AGENT4WEAKNESS. First, we evaluate the reports generated by AGENT4WEAKNESS across four dimensions, including Requirement Fulfillment, Content Value, Factuality, and Readability. The reports show a significant improvement of 2.6 points out of 10 when evaluated by LLMs, compared to the baseline. Through human studies, we find that LLM-assigned scores align well with human ratings. This result confirms the high quality of the reports generated by AGENT4WEAKNESS. In addition, AGENT4WEAKNESS achieves an improvement of 3.4 over the baseline in the Content Value dimension, demonstrating that our method can generate rich analyses of evaluation disparities, providing a sufficient comparison. Furthermore, AGENT4WEAKNESS scores an improvement of 3.4 compared to the baseline in the Requirement Fulfillment dimension, indicating that our method ensures the reports meet user requirements, showcasing its flexibility in evaluation. Additional experiments reveal that model performance improves by 3.7 when guided by the weakness discovered from AGENT4WEAKNESS. Notably, this gain is achieved purely through prompt guidance rather than training, further validating that such reports can effectively drive performance improvements and highlighting the potential for practical applications of our method.

Our contributions are as follows:

- To address the shortcomings of insufficient comparison and inflexible evaluation in existing weakness discovery methods, we propose AGENT4WEAKNESS, which leverages multi-agent collaboration to ensure that the generated reports are both sufficient and flexible.

- Experimental results on 8 models and 27 datasets demonstrate that the reports generated by AGENT4WEAKNESS achieve the improvement of 2.6 points out of 10 compared with the baseline

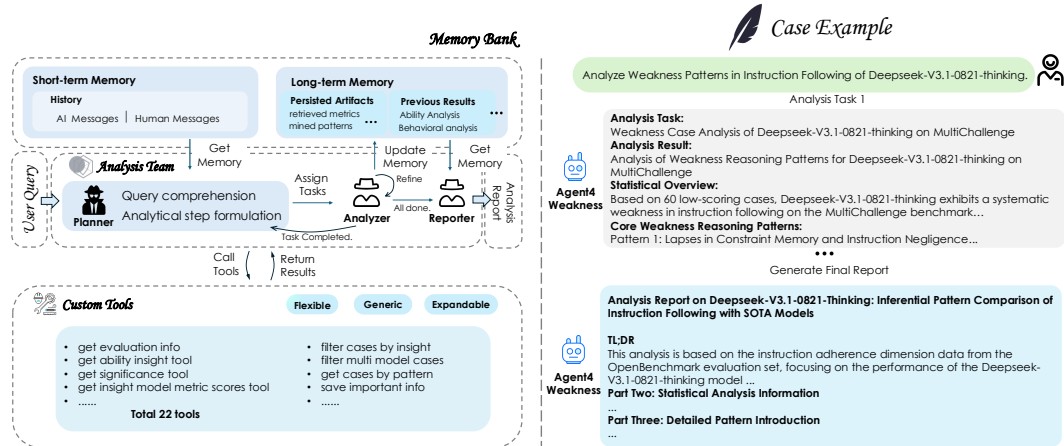

Figure 2: The left side shows the overview of AGENT4WEAKNESS, including the Planner, Analyzer, and Reporter, which perform capability and behavior analysis. The right side presents a concrete example of capability weakness analysis for the models.

across four evaluation dimensions and the high consistency between model and human scores, confirming the high quality of the reports produced by our method.

- Guided by the reports generated by AGENT4WEAKNESS, model performance improves by an average of $3.7$ using the discovered weakness, demonstrating the practical value of our method.

## 2 METHODOLOGY

At a high level, our method builds an agent-based system that leverages model evaluation data to answer user queries. We first collect and organize diverse evaluation results from a wide range of models and benchmarks, covering multiple capability dimensions. On top of this evaluation corpus, we design a multi-agent workflow, equipping agents with specialized tools that allow them to retrieve, process, and analyze the evaluation data. When a user issues a query, the system coordinates these agents to plan the analysis steps, invoke the appropriate tools for evidence gathering, and compose a structured response. In this way, the method provides not only quantitative comparisons across models but also interpretable explanations of their weaknesses and behavioral tendencies.

### 2.1 OVERVIEW

**Task Definition.** Formally, given a user query $q \in \mathcal{Q}$ and evaluation data $\mathcal{D}_{\text{eval}}$, our system AGENT4WEAKNESS produces a structured report $y \in \mathcal{Y}$ that integrates quantitative evidence with typical failure traces:

$$y = \text{AGENT4WEAKNESS}(q, \mathcal{D}_{\text{eval}}). \tag{2.1}$$

**Data and Query.** We evaluate a set of $104$ models, denoted by $\mathcal{M}$, on 27 distinct benchmarks, denoted by $\mathcal{B}$. These benchmarks are organized into seven capability dimensions: *Comprehensive*, *Reasoning*, *Math*, *Code*, *Instruction Following*, *Knowledge & Hallucination*, and *Multilingual* (details in Appendix C). Given a user query $q$ and the comprehensive evaluation data $\mathcal{D}_{\text{eval}}$, AGENT4WEAKNESS produces a report $y$. We formally define the evaluation data as a tuple $\mathcal{D}_{\text{eval}} = (\mathcal{D}_{\text{raw}}, \mathcal{D}_{\text{stat}})$. (i) $\mathcal{D}_{\text{raw}}$ represents the instance-level data. Let $\mathcal{I}_b$ be the set of instance indices for a given benchmark $b \in \mathcal{B}$. $\mathcal{D}_{\text{raw}}$ is the aggregation of all instance-level results across all benchmarks:

$$\mathcal{D}_{\text{raw}} = \bigcup_{b \in \mathcal{B}} \{(q_i, a_i^*, \{r_{m,i}\}_{m \in \mathcal{M}}) \mid i \in \mathcal{I}_b\} \tag{2.2}$$

where $q_i$ is an instance from benchmark $b$, $a_i^*$ is its corresponding ground-truth answer, and $\{r_{m,i}\}_{m \in \mathcal{M}}$ is the set of raw responses from all models in $\mathcal{M}$ for that instance. (ii) $\mathcal{D}_{\text{stat}}$ comprises derived statistics aggregated at the model and benchmark levels. Let $K$ be the number of distinct

statistic types computed (e.g., accuracy, runtime, Best-of-$N$). $\mathcal{D}_{\text{stat}}$ is the set of all pre-computed metrics for every model-benchmark pair:

$$\mathcal{D}_{\text{stat}} = \{\text{stat}_k(m, b) \mid m \in \mathcal{M}, b \in \mathcal{B}, k \in \{1, \ldots, K\}\} \tag{2.3}$$

where $\text{stat}_k(m, b)$ represents the resulting value of the $k$-th statistic for model $m$ on benchmark $b$.

**Agent Workflow.** The system consists of three agents: **Planner** $\mathcal{P}$, **Analyzer** $\mathcal{A}$, and **Reporter** $\mathcal{R}$. Each agent is specified by (*i*) *role* (input→output mapping), (*ii*) *tools* (only $\mathcal{A}$ uses external tools), and (*iii*) *memory* with long-term $\mathbf{K}$ (persisted artifacts such as retrieved metrics, mined patterns, and reusable templates) and short-term $\mathbf{H}$ (within-run conversational/plan state). All prompts embed background and priors about the evaluation setting, and each agent receives its own prompt together with the cross-agent history $\mathcal{H}$ for grounding and consistency (Appendix E). Overall, the Planner turns the user query into an analysis plan, the Analyzers derive weakness observations, and the Reporter composes the final report. This coordination follows a structured data flow: (1) The Planner design the plan $\pi$ according to $q$ and $\mathcal{D}_{\text{eval}}$ (Equation 2.4). (2) The Analyzer executes each sub-task in $\pi$ by querying $\mathcal{D}_{\text{eval}}$ via tools, generating a set of observations $\{o_i\}$ (Equation 2.5). (3) The Reporter aggregates $\{o_i\}$ to compose the final report $y$ (Equation 2.6).

**Two Complementary Analyses Tasks.** Based on the granularity of available data, we decompose the task into two levels. *Capability analysis* performs numerical calculations to discover weaknesses, such as estimating per-dimension performance vectors, computing gaps, and assessing significance and tiering. *Behavioral analysis* mines repeated reasoning patterns from raw responses by contrasting low-scoring cases of the model with high-scoring cases from other models.

## 2.2 PLANNER

**Role and Memory.** The Planner interprets $q$ and $\mathcal{D}_{\text{eval}}$ in context, aligns them with background priors, and constructs a weakness-oriented analysis plan $\pi$. Formally, this plan is a sequence of analysis steps $\pi = (s_1, \ldots, s_k)$, where each step $s_j = (\mathcal{T}_j, \text{args}_j)$ specifies some tools $\mathcal{T}_j$ and $\text{args}_j$ denotes their arguments, explicitly defining the slices and tool uses for analysis. It also decides whether to continue or to replan when Analyzer observations is underperforming. Long-term memory $\mathbf{K}_{\mathcal{P}}$ stores reusable plan schemas, subgoal taxonomies, and adequacy thresholds per dimension; short-term memory $\mathbf{H}_{\mathcal{P}}$ maintains the current plan, assigned slices, and the latest observations to enable iterative refinement. Formally, with cross-agent history up to step $t-1$ denoted by $\mathcal{H}_{1:t-1}$,

$$(\pi, d, \mathbf{H}'_{\mathcal{P}}) = \mathcal{P}(q, \mathcal{D}_{\text{eval}}, \mathbf{K}_{\mathcal{P}}, \mathbf{H}_{\mathcal{P}}, \mathcal{H}_{1:t-1}), \quad d \in \{\texttt{continue}, \texttt{replan}\}. \tag{2.4}$$

The Planner's prompt includes demonstrations that translate user intent into capability- and behavior-oriented subgoals, enumerate target slices, and schedule analyzers to derive progressive observations with principled stopping criteria, ensuring the flexible requirement fulfillment.

Table 1: Examples of the tools used in AGENT4WEAKNESS include their names, purposes, and categories, with detailed tool information provided in Appendix F.

| Tool | Purpose | Category |
|------|---------|----------|
| get ability tool | Return a Markdown table listing scores of multiple models across capability dimensions and benchmarks. | Data acquisition |
| get significance tool | Using the specified model as the baseline, compute other models' score differences, percentage changes, improvements, and statistical significance relative to the baseline. | Data analysis |
| get cases by pattern | Automatically analyze all cases of the specified benchmark. | In-depth analysis |

## 2.3 ANALYZER

**Role, Tool, and Memory.** The Analyzer executes the plan $\pi = (s_1, \ldots, s_k)$ over $\mathcal{D}_{\text{eval}}$. It iterates through the plan, invoking the specified tools for each step $s_j = (\mathcal{T}_j, \text{args}_j)$ to process data. The collective result is a set of observations $o = \{o_j\}_{j=1}^{k}$ that are of high factuality and content value.

It uses three families of tools: (*i*) *Data acquisition* $\mathcal{T}_{\text{daq}}$ for retrieving benchmark summaries, per-slice statistics (means/variances, Best/Worst-of-$N$), usage signals (token, runtime), and filtered case sets under predicates (error type, length, timeouts, etc.); (*ii*) *Statistical analysis* $\mathcal{T}_{\text{stat}}$ for computing paired gaps against references, ranking and tiering, bootstrap confidence intervals, effect sizes, and capability/benchmark correlations; (*iii*) *In-depth analysis* $\mathcal{T}_{\text{deep}}$ for scalable pattern mining over raw responses using a fast model-as-tool to summarize repeated behaviors (e.g., premature finalization, tool-call misfires, brittle chain-of-thought). We select one tool from each category for presentation in Table 1. Long-term memory $\mathbf{K}_{\mathcal{A}}$ persists retrieved evidence, indices to representative cases, and mined pattern schemas; short-term memory $\mathbf{H}_{\mathcal{A}}$ caches the current tool-call trace and samples for rapid within-run access. The Analyzer's input–output is

$$o = \mathcal{A}(\pi, \mathcal{D}_{\text{eval}}, \mathbf{K}_{\mathcal{A}}, \mathbf{H}_{\mathcal{A}}; \mathcal{T}_{\text{daq}}, \mathcal{T}_{\text{stat}}, \mathcal{T}_{\text{deep}}). \tag{2.5}$$

**Capability Analysis.** When diagnosing a capability weakness in the set of capability dimensions $c \in \mathcal{C}$, the Analyzer primarily invokes $\mathcal{T}_{\text{daq}}$ and $\mathcal{T}_{\text{stat}}$ to compute evidence, such as performance vectors and reference summaries, gaps with uncertainty (e.g., bootstrap CIs, $p$-values), and tier assignment. This yields quantitative evidence used to localize and prioritize capability deficits.

**Behavioral Analysis.** When diagnosing a behavioral weakness, the Analyzer constructs a contrast set by pairing low-scoring cases from $m^{\star}$ with high-scoring matched cases from $R$ by $\mathcal{T}_{\text{daq}}$. It then applies $\mathcal{T}_{\text{deep}}$ to mine frequent weakness patterns and validate them against raw traces, producing evidence, such as contrastive summaries (error type, trigger, missing step), pattern set $\mathcal{P}_{\text{beh}}$ with prevalence and representative exemplars, and hypotheses linking patterns to capability gaps (e.g., *hallucination under tool-latency $\rightarrow$ knowledge & control*). This yields qualitative evidence that explains *why* the evaluation scores are low.

## 2.4 REPORTER

**Role and Memory.** The Reporter turns a multiset of analyzer outputs $\{o_i\}$ into a concise report $y$ that ranks weaknesses by impact, pairs each claim with quantitative evidence (metrics/tables) and qualitative traces (typical failure cases), and preserves clarity across the seven dimensions. Long-term memory $\mathbf{K}_{\mathcal{R}}$ stores templates and stable claim↔evidence linking patterns into $\mathbf{K}_{\mathcal{A}}$; short-term memory $\mathbf{H}_{\mathcal{R}}$ maintains the evolving outline and pending citations to ensure coherence and completeness. Its input–output mapping is

$$y = \mathcal{R}(\{o_i\}_{i=1}^{m}, \mathbf{K}_{\mathcal{R}}, \mathbf{H}_{\mathcal{R}}, \mathcal{H}_{1:t}). \tag{2.6}$$

Practically, the Reporter performs evidence binding (claims $\rightarrow$ linked metrics and cases), resolves redundancies across overlapping slices, and enforces style constraints (headlines, captions, and citation format) specified in the prompt, yielding a final evaluation report that is readable yet fully traceable to the underlying computations.

## 3 EXPERIMENTS

### 3.1 SETTINGS

We first evaluate several models on a range of benchmarks, recording a comprehensive set of statistical metrics and the corresponding model responses. A detailed list of all 104 models and 27 benchmarks is provided in Appendix C.

**Models.** We employ Claude-Opus-4.1-thinking (Anthropic, 2025) to run AGENT4WEAKNESS and query the 8 models individually regarding their weakness, including: GPT-5-high (OpenAI, 2025), Grok-4 (xAI, 2025), Claude-Opus-4.1-thinking (Anthropic, 2025), Gemini-2.5-pro (Google, 2025b), Qwen-3-235B-A22B-Thinking-2507 (Qwen Team, 2025), Seed-1.6-Thinking-250715 (ByteDance, 2025), Deepseek-V3.1-0821-Thinking (DeepSeek-AI et al., 2025), and Gemini-2.5-Flash-0520 (Google, 2025a), which are all mainstream LLMs currently. We also present the results of AGENT4WEAKNESS using GPT-5 and Gemini-2.5-pro in Appendix G.2.

**Queries.** We conduct the following inquiries for each model, including: *Q1*. Analyze the weaknesses in the model's capabilities. *Q2*. Analyze the disadvantages of the model in terms of time cost

Table 2: Definitions of four dimensions and the deduction rules on a 10 point scale.

| Criteria | Primary Definition | Deduction Logic |
|---|---|---|
| **Requirement Fulfillment** | Evaluates the agent adherence to both general and specific instructions within query. | • Non-adherence: $-1$ points |
| **Content Value** | Assesses the utility of the output, including structural integrity and the soundness of the analysis. | • Incomplete structure: $-1$ to $-3$ points
• Missing a primary category: $-3$ points
• Missing a secondary category: $-1$ to $-2$ points
• Incomplete case presentation table: $-2$ points
• Unsound analysis: $-1$ to $-2$ points
• Inappropriate primary category: $-2$ points
• Inappropriate secondary category: $-1$ point |
| **Factuality** | Verifies the accuracy and reliability of data citations and external links. | • A single instance of a factual error: $-2$ points |
| **Readability** | Measures the clarity, fluency of the language, and the effectiveness of case presentation. | • Expressive or logical flaws: $-0.5$ points per instance
• Poor reading experience: $-1$ point |

and token consumption. *Q3*. Analyze the weaknesses in the model's behavior. *Q4*. Does the model instruction non-compliance exist, and under what circumstances is this phenomenon most severe? *Q5*. Analyze the deficiencies exhibited by the model in reflective behavior. *Q6*. Analyze the relationship between the model's abilities and its maximum ability limit. Q1-Q3 are general inquiries addressing performance limitations, resource consumption, and behavioral deficits, respectively, in contrast to the more specific questions Q4-Q6. These queries are designed based on key concerns collected from LLM practitioners (see Appendix G.1 for a detailed discussion).

**Baselines.** To highlight the quality of the analysis generated by AGENT4WEAKNESS, we establish two baselines. (*i*) Direct question-answering (Direct QA) baseline involves feeding all relevant performance weakness data into the model and prompting it to answer the query directly. (*ii*) One-Agent baseline uses the same tools as AGENT4WEAKNESS, but without specialized roles. We do not compare with prior works because existing methods rely on fixed pipelines that cannot flexibly accommodate all of our queries. Moreover, prior studies analyze each model in isolation without incorporating evaluation data from other models, which would render our comparison unfair. We compare the efficiency of with baselines in Appendix G.3.

**Evaluation.** To assess the quality of the analysis generated by AGENT4WEAKNESS, we conduct both human and model-based evaluations. Specially, we employ professional evaluators to score the generated analyses across four dimensions, each on a scale from 0 to 10. The detailed scoring rubric is shown in Table 2. Furthermore, we also use Claude-Opus-4.1-thinking to assign scores following the same rubrics, and we report the average score over 5 runs. We discuss the robustness of AGENT4WEAKNESS in Appendix G.4.

## 3.2 MAIN RESULTS

Our experimental results are shown in Table 3, where the reported scores are LLM ratings. AGENT4WEAKNESS consistently outperforms baselines, achieving average improvements of 2.7, 3.4, 3.4, and 1.5 on *Requirement Fulfillment*, *Content Value*, *Factuality*, and *Readability*, respectively. These results demonstrate that AGENT4WEAKNESS not only enables thorough model comparison and highlights content value while flexibly satisfying user needs, but also produces weakness analyses with high factual accuracy and readability.

**Finding 1. Performance gains are particularly pronounced on complex queries.** The improvements of AGENT4WEAKNESS are larger on Q2 and Q3 than on Q1. Q1 primarily involves pairwise performance comparisons, whereas Q2 requires synthesizing multiple factors such as runtime and token usage, and Q3 demands deeper reasoning analysis across benchmarks. The baseline struggles with these more complex cases, whereas AGENT4WEAKNESS demonstrates robust performance.

Table 3: Comparison of model evaluations for the baseline and AGENT4WEAKNESS across 4 evaluation dimensions and 6 queries, with a maximum score of 10. Avg denotes the average scores across the 6 queries on the same dimensions. The highest average score is highlighted in **bold**.

| Method | Query | Requirement Fulfillment | Content Value | Factuality | Readability |
|---|---|---|---|---|---|
| Direct QA | Q1 | 6.7 | 5.7 | 6.4 | 7.9 |
| | Q2 | 3.7 | 3.0 | 3.3 | 6.4 |
| | Q3 | 6.9 | 6.6 | 7.3 | 8.3 |
| | Q4 | 5.0 | 3.0 | 4.3 | 7.3 |
| | Q5 | 4.3 | 2.8 | 3.0 | 5.3 |
| | Q6 | 7.5 | 6.0 | 4.5 | 6.5 |
| | Avg | 5.7 | 4.5 | 4.8 | 6.7 |
| One-Agent | Q1 | 6.7 | 5.7 | 6.0 | 7.3 |
| | Q2 | 6.7 | 5.5 | 5.0 | 6.4 |
| | Q3 | 7.0 | 4.5 | 6.4 | 5.0 |
| | Q4 | 4.7 | 5.3 | 5.3 | 6.4 |
| | Q5 | 7.0 | 5.7 | 5.0 | 6.7 |
| | Q6 | 6.7 | 7.0 | 6.5 | 8.0 |
| | Avg | 6.5 | 5.5 | 5.7 | 6.6 |
| AGENT4WEAKNESS | Q1 | 8.8 | 8.3 | 9.7 | 7.5 |
| | Q2 | 9.9 | 8.6 | 8.0 | 8.4 |
| | Q3 | 8.9 | 8.7 | 8.1 | 8.7 |
| | Q4 | 8.7 | 7.9 | 8.4 | 7.7 |
| | Q5 | 8.2 | 8.1 | 8.2 | 8.2 |
| | Q6 | 8.5 | 8.5 | 9.2 | 8.3 |
| | Avg | **8.8** | **8.4** | **8.6** | **8.1** |

**Finding 2. The significant benefits appear in *Requirement Fulfillment* and *Content Value*, and *Factuality*.** While the baseline can produce shallow comparisons, excessive input length harms instruction adherence and the absence of specialized tools limits its analytical depth (Liu et al., 2024; Wu et al., 2025). By contrast, AGENT4WEAKNESS generates substantially richer and more faithful reports.

**Finding 3. The gain in *Readability* is modest but consistent.** Although AGENT4WEAKNESS outperforms the baselines, improvements are smaller than in other dimensions. This is because models inherently exhibit their own stylistic tendencies, such as habitual word choices and preferred rhetorical structures. Even with explicit guidance on report style, it is challenging to achieve significant gains in readability (Wang et al., 2025a).

**Finding 4. Model-based scores align strongly with human evaluation.** To validate the reliability of model ratings, we collect human annotations, with details in Appendix D. The results show strong positive correlations: Pearson $r = 0.801$, 95% CI = $[0.12, 0.97]$, $t(5) = 2.99$, $p \approx 0.03$; Spearman $\rho = 0.944$, $p < 0.01$ ($n = 7$). This indicates very high rank-order alignment and substantial linear agreement, with only minor deviations in magnitude across individual items.

## 4  DISCUSSION

### 4.1  RQ1. CAN OUR METHOD ACCURATELY IDENTIFY MODEL WEAKNESSES?

To validate whether AGENT4WEAKNESS accurately identifies model weaknesses, we conduct verifiable analyses. For capability analysis, we input the scores of 106 models on 51 benchmarks and ask both AGENT4WEAKNESS and the baseline to identify the benchmark where a given model ranks the lowest, and the model that ranks the lowest within a specified capability dimension. The target models are consistent with those in the main experiments, and the capability dimensions include overall, reasoning, math, code, instruction following, knowledge, and multilingual capabilities. To ensure fairness, AGENT4WEAKNESS does not include tools that directly return these answers; instead, the agent must retrieve the relevant data, compute results, or verify them using auxiliary tools.

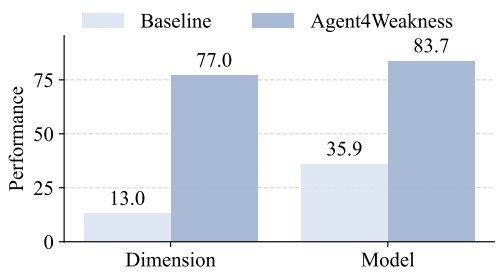

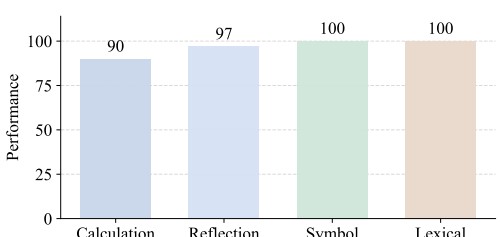

Figure 3: Accuracy in identifying weakness capability dimensions and underperforming models, compared with baselines.

Figure 4: Accuracy of AGENT4WEAKNESS in detecting weaknesses across four behavioral patterns.

As shown in Figure 3, AGENT4WEAKNESS improves accuracy by an average of 55.9 points over the baseline, demonstrating its effectiveness. We observe that errors mainly arise from hallucinations due to overly long contexts after multiple tool calls, while the baseline struggles to identify the weakest models or dimensions from large-scale data (Liu et al., 2024; Wu et al., 2025).

For behavioral analysis, we instruct AGENT4WEAKNESS to detect quantifiable patterns in the outputs of a specified model, including calculation errors, reflection mechanisms, symbol preferences, and lexical preferences. As shown in Figure 4, AGENT4WEAKNESS accurately identifies these behavioral features and computes their frequencies, confirming its ability to detect both capability and behavioral weaknesses in models. The accuracy of detecting calculation errors is the lowest because the model needs to call external tools or perform calculations on its own. This challenge is particularly evident in high-difficulty mathematical benchmarks such as AIME (AIME, 2025) and OlympiadBench (He et al., 2024), where identifying a miscalculation at a specific step is more difficult than recognizing symbols or words.

## 4.2 RQ2. ARE THE ROLES AND TOOLS EMPLOYED IN AGENT4WEAKNESS EFFECTIVE?

To assess the contribution of roles and tools in AGENT4WEAKNESS, we conduct ablation studies (Table 4). For the role input, we keep only the task-specific instructions and the preceding agents' context in the role prompt, removing background about the evaluation and any prior knowledge. For tools, AGENT4WEAKNESS uses data acquisition, data analysis, and deep analysis tools. In the experiments of ablating analysis tools, we retain only data acquisition, and in the experiments of ablating in-depth tools, we only remove the deep analysis tools. Furthermore, we individually ablate the Planner and Reporter agent. Since the Analyzer is essential for acquiring data, it cannot be fully ablated. Therefore, the aforementioned ablations of the analysis and in-depth tools can be regarded as a partial ablation of the Analyzer.

Based on the experimental results in Table 4, we can observe that: (*i*) When ablating background knowledge, scores on *Requirement Fulfillment* and *Readability* also decline, as the agents no longer sufficiently understand the query or the provided data. The resulting reports are less structured and more hyperbolic, further reducing readability. (*ii*) When ablating tools, *Content Value* and *Factual Accuracy* degrade the most, because tools help verify data fidelity and enable more thorough comparisons and deeper analyses of model weaknesses. In the experiment ablating the in-depth tools, performance on Q1 and Q2 remains consistent with the full AGENT4WEAKNESS model. This is expected, as these specific tools are designed to analyze reasoning patterns, which are not utilized in the responses for Q1 and Q2. (*iii*) Comparatively, the performance degradation from ablating Requirement Fulfillment is not significant for Q1 and Q2, but it is substantial for Q3. This suggests that while the model is inherently inclined to identify information about capabilities and output costs, it does not proactively analyze its own behavior. Such behavioral analysis, in turn, necessitates a more fine-grained examination of the evaluation data. (*iv*) The Planner agent contributes most significantly to Content Value. It strategically plans the analysis from multiple perspectives, which facilitates the discovery of a more diverse set of model deficiencies. (*v*) Ablating the Reporter not only severely degrades Readability but also causes a decline in other metrics. This demonstrates that the Reporter is not a simple copy-paste mechanism but a critical component responsible for synthesizing, refining, and validating the final output.

Table 4: Ablation study across three queries (Q1–Q3), with a maximum score of 10. Metrics are Requirement Fulfillment (R), Content Value (C), Factuality (F), and Readability (R).

| Method | Q1 | | | | Q2 | | | | Q3 | | | |
|---|---|---|---|---|---|---|---|---|---|---|---|---|
| | R | C | F | R | R | C | F | R | R | C | F | R |
| Agent4Weakness | **8.8** | **8.3** | **9.7** | **7.5** | **9.9** | **8.6** | **8.0** | **8.4** | **8.9** | **8.7** | **8.1** | **8.7** |
| Ablating roles | 8.3 | 7.4 | 8.6 | 6.7 | 7.9 | 8.0 | 7.7 | 6.1 | 4.6 | 1.6 | 1.6 | 6.1 |
| Ablating analysis tools | 7.9 | 7.0 | 8.3 | 7.4 | 9.3 | 8.4 | 7.9 | 7.7 | 5.3 | 4.3 | 2.0 | 6.0 |
| Ablating in-depth tools | 8.8 | 8.3 | 9.7 | 7.5 | 9.9 | 8.6 | 8.0 | 8.4 | 6.0 | 4.5 | 4.2 | 7.5 |
| Ablating Planner | 6.5 | 6.1 | 6.5 | 7.4 | 8.0 | 6.4 | 7.3 | 6.0 | 5.6 | 2.0 | 6.0 | 6.2 |
| Ablating Reporter | 7.2 | 6.5 | 6.8 | 6.3 | 6.7 | 5.0 | 6.6 | 5.8 | 4.8 | 3.0 | 5.0 | 5.7 |

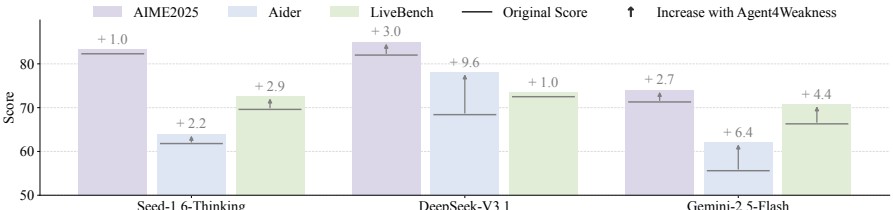

Figure 5: The original model scores versus the scores after implementing the weakness analysis and improvement suggestions provided by AGENT4WEAKNESS.

### 4.3 RQ3. CAN ANALYSIS GENERATED BY AGENT4WEAKNESS IMPROVE MODEL PERFORMANCE?

To evaluate the accuracy and effectiveness of AGENT4WEAKNESS, we feed its analysis of a model's evaluation results back into the same model to determine if the analysis improves performance. Specifically, we input the evaluation results of models on AIME2025 (AIME, 2025), Aider (Gauthier, 2025), and the LiveBench 2025-04-25 version (White et al., 2025) into AGENT4WEAKNESS, respectively. Subsequently, we provide the behavioral weaknesses and improvement suggestions identified by AGENT4WEAKNESS to the same model and observe its performance change. As shown in Figure 5, AGENT4WEAKNESS consistently improves the model performance by an average of 3.7 points through prompt modifications, demonstrating the effectiveness of our analysis.

Specifically, we have the following key findings: (*i*) The performance of the model improves because targeted prompts about potential weaknesses in reasoning patterns and corresponding suggestions for improvement enhance its reasoning behavior. For instance, AGENT4WEAKNESS identifies that the reasoning process of DeepSeek-V3.1 on AIME2025 questions is disorganized. By contrasting and summarizing the reasoning processes from the model's successful cases, AGENT4WEAKNESS therefore suggests using markers such as "### Step 1" to structure the reasoning and adding verification of intermediate results after each step. After receiving this prompt, the model exhibits a more organized reasoning process and consistently adopts the reasoning markers. Furthermore, AGENT4WEAKNESS finds that DeepSeek-V3.1 applies congruence properties superficially, without considering the structures of group theory and ring theory. It thus recommends fully utilizing modular arithmetic, the Chinese Remainder Theorem, and Euler's theorem, and conducting a deeper analysis of the group-theoretic structures and algebraic properties of higher-order congruences. The prompted model then applies these theorems in its reasoning to solve problems successfully, leading to a performance increase. (*ii*) The improvement on Seed-1.6-Thinking is not significant. This is because Seed-1.6-Thinking has a weaker capability for instruction following in our experiments, which prevents it from adhering well to the corrective suggestions. (*iii*) The most substantial improvement is observed on Aider. For code tasks, the correct action to fix an error is highly specific. Therefore, when an error is prompted to the model, it can learn a very concrete and executable rule. For example, our method finds that Gemini-2.5-Flash tends to engage in preemptive error analysis, introduce general debugging methods, or discuss hypothetical problems, which results in incorrect or excessively long error analyses. It therefore suggests that error analysis should occur only when correcting the code and should focus on the direct cause of the current, specific problem.

## 5 RELATED WORK

As benchmark scores are insufficient for revealing fine-grained and in-depth model weaknesses, prior research focuses on identifying these granular deficiencies from detailed evaluation results (Moayeri et al., 2025; Brown et al., 2025; Wang et al., 2025b).

### 5.1 CAPABILITY WEAKNESS DISCOVERY

Existing methods identify model weaknesses by extracting the capabilities required to answer questions and organizing them into sets or capability trees. For example, QualEval (Murahari et al., 2024) utilizes a LLM to summarize potential taxonomies from samples and then maps all benchmark questions to specific categories. Thus, QualEval can identify the domains and skills that correspond to lower performance as weaknesses. Similarly, EvalTree (Zeng et al., 2025) constructs hierarchical capability trees to profile model performance. Each node in the tree represents a specific capability, and the structure facilitates the identification of performance deficits at various levels of granularity. This method enables a more detailed understanding of model weaknesses across different capabilities. SkillVerse (Tian et al., 2025) introduces a tree-structured framework for assessing model proficiency. By organizing capabilities into a hierarchical structure, SkillVerse allows for a nuanced analysis of model strengths and weaknesses, guiding targeted improvements.

However, existing works are limited to simple performance comparisons, failing to meticulously consider non-performance metrics Hu & Zhou (2024). Furthermore, these methods overlook referential information from other models, which results in conclusions with low confidence (Luettgau et al., 2025). Moreover, existing works employ fixed pipelines, thereby confining the analysis of model weaknesses to limited aspects (Brawer et al., 2023). In contrast, AGENT4WEAKNESS is implemented as a multi agent system that incorporates professional statistical tools to achieve comprehensive comparisons and flexible evaluation.

### 5.2 BEHAVIORAL WEAKNESSES DISCOVERY

Previous works analyze deficiencies in model behavior to better understand the models and guide their improvement (Chang & Bergen, 2024). ReportCards (Yang et al., 2024) provides human-interpretable, natural language summaries of model behavior, focusing on specific skills or topics. This qualitative approach facilitates the identification of behavioral patterns that may indicate underlying weaknesses. CoT Encyclopedia (Lee et al., 2025) employs a clustering technique to group evaluation data based on observed patterns, deriving scores for different model capabilities. This method allows for the identification of behavioral trends across various tasks, contributing to a deeper understanding of model performance.

However, their methods are fixed and lack interactivity, making them unable to flexibly meet user needs. Additionally, they do not incorporate other models in their weakness analysis. For example, a model may make errors on highly challenging questions, but even the best models are beyond their cognitive limits on such questions, meaning that these weaknesses are not the primary behavioral weaknesses of the analyzed model.

## 6 CONCLUSION

We introduce AGENT4WEAKNESS, a tool-augmented multi-agent framework that converts raw evaluation data into targeted weakness reports with both sufficient comparison and flexible evaluation. At the evaluation of 104 models and 27 benchmarks, AGENT4WEAKNESS improves LLM scores by 2.6/10 on average and exhibits strong agreement with human raters. A 3.4 score improvement in Content Value proves that AGENT4WEAKNESS has high sufficient evaluation. Besides, AGENT4WEAKNESS surpasses a strong baseline by a 3.4 score improvement in Requirement Fulfillment, showing that our method can generate reports with flexible evaluation. Ablation results further confirm that explicit role design and targeted tool use are key drivers of these gains. Further evaluation experiments show that using the report generated with AGENT4WEAKNESS, performance improves 3.7 compared with baselines, showing the practical value of our method. By producing evidence-grounded findings that localize weaknesses, AGENT4WEAKNESS provides a practical foundation for weakness discovery in the LLM era.

## 7 REPRODUCIBILITY

We have provided all prompts of this paper in Appendix E. We release the code in `https://anonymous.4open.science/r/agent_code`.

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

## A  THE USE OF LARGE LANGUAGE MODELS (LLMS)

We used LLMs only to polish our paper for better clarity and fluency, without involving the core research content. All contents were checked and edited by the authors to ensure the quality and alignment. The authors take full responsibility for the final version of the paper.

## B  ETHICS

All models used in this paper are publicly available, and our usage follows their licenses and terms. Additionally, we confirm that the compensation provided to evaluators is significantly higher than the local minimum wage.

## C  EVALUATION BACKGROUND

### C.1  MODELS

We evaluate the following models: qwen-3-next-80b-a3b-thinking, qwen-3-next-80b-a3b-instruct, qwen3-max-preview, GPT-5-high, qwen-3-4b (think), GPT-OSS-20b-medium, Hunyuan-T1-0711, qwen-3-coder-plus, qwen-3-235b-a22b-instruct-2507 (nothink), qwen-3-235b-a22b-thinking-2507, Grok-4, Grok-3, Llama-4-Maverick, Llama-4-Scout, Grok-3-mini-high, ChatGPT-4o-latest, Doubao-1.5-Lite-32k.250115, Doubao-1.5-Pro-32k.250115, Doubao-1.5-Thinking-Pro-M.250415, Doubao-1.5-Thinking-Pro.250415, Seed-1.6-Flash.250615, Seed-1.6-Thinking.250615, Seed-1.6-AutoCoT.250615-AutoCoT, Seed-1.6-AutoCoT.250615-NoCoT, DeepSeek-R1-0528, o4-mini-high, qwen-plus-0428 (nothink), GPT-4.1-nano, DeepSeek-V3-0324, Gemini-2.0-Flash, Claude-4-Sonnet-nothinking, GPT-4o-1120, o3-mini-high, Minimax-Text-01, GPT-4.1-mini, Baichuan4-Turbo, o1-high, Gemini-2.0-Flash-Lite, qwen-max-0125 (nothink), GLM-4-Air.0414, Mistral-large-2411, Nova-pro, Yi-lightning, Claude-3.7-Sonnet, GPT-4.1, Gemini-2.5-flash.0520, qwen-3-235b-a22b-2504 (think), qwen-turbo-0428 (nothink), SenseNova-V6-Turbo, SenseNova-V6-Pro, Gemini-2.5-pro.0605, ERNIE-4.5-Turbo-32K, GLM-Z1-Air.0414, Claude-3.7-Sonnet-thinking, Claude-4-Sonnet-thinking, qwen-3-30b-a3b (think), qwen-3-32b (think), ERNIE-X1-Turbo-32K, 360-gpt2-o1, SenseNova-V6-Reasoner, Claude-4-Opus-nothinking, StepFun-2-16k, Claude-4-Opus-thinking, StepFun-R1-V-mini, Kimi-Thinking-preview, Gemini-2.5-Pro, Gemini-2.5-Flash, dots-llm1, Gemini-2.0-flash-lite-preview.0617, Hunyuan-T1-0529, GPT-4o-mini, ERNIE-4.5-Turbo-128K-Preview.0629, ERNIE-4.5-300b-a47b, o3-high, qwen-plus-0714 (nothink), qwen-turbo-0715 (nothink), Kimi-K2, qwen-3-coder-480b-a35b-instruct, Gemini-2.5-Flash-Lite, qwen-3-30b-a3b-instruct-2507 (nothink), qwen-3-30b-a3b-thinking-2507, Seed-1.6-Thinking-agent-preview, GLM-4.5, GLM-4.5-AirX, GLM-4.5-Air, GPT-OSS-20b-high, GPT-5-mini-high, GPT-5-nano-high, GPT-5-chat, GLM-4.5-X, GPT-5-medium, Claude-Opus-4.1-nothinking, 360-zhinao2-o1.5, StepFun-3, Claude-Opus-4.1-thinking, Deepseek-V3.1-0821-nothinking, Deepseek-V3.1-0821-thinking, Seed-1.6-AutoCoT.250615-CoT, Seed-1.6-Flash.250715, Seed-1.6-Thinking.250715, Kimi-K2-0905, GPT-OSS-120B-low, GPT-OSS-120B-medium, and GPT-OSS-120B-high.

### C.2  BENCHMARKS

We consider the following benchmarks: MMLU pro, MMLU, Humanity Last Exam, GPQA diamond, SuperGPQA, LiveBench, MixEvalHard, ArenaHard, ARCAGI, ProcBench, KORBench, ZebraLogicBench, AIME 2025, AIME 2024, HARP, Omni MATH, OlympiadBench, SWE Bench Verified, SWE Lancer, Aider, LiveCodeBench, MultiChallenge, IFEval, Collie Hard, Chinese SimpleQA, SimpleQA, and MMMLU.

## D  HUMAN EVALUATIONS

We present the average human and model scores on Q1, as shown in Table 5. The results indicate a high consistency between human and model evaluations, with fluctuations not exceeding 0.3.

Table 5: Comparison of model scores and human scores on Q1.

|  | Requirement Fulfillment | Content Value | Factuality | Readability |
|---|---|---|---|---|
| Human | 8.8 | 8.0 | 9.5 | 7.5 |
| Model | 8.8 | 8.3 | 9.7 | 7.5 |

# E PROMPTS

**Planner (Capability Analysis)**

Background

You are working inside the **PostEvalAgent** system.

1. **What is "PostEvalAgent"?**
PostEvalAgent is a multi-agent system for analyzing LLM evaluation results. It helps us better understand the data produced by evaluations, thereby improving our understanding of models and guiding optimization.

2. **A quick introduction to the evaluated data**
To clarify the task, we briefly introduce the evaluation data, which has several layers:
- **case (smallest unit):** Contains fields such as 'prompt', 'response', 'ground truth', 'metric_name', 'score', 'tag', etc.; uniquely identified by a global '__internal_id__'.
- **exercise:** An aggregation of multiple cases; it may correspond to the full set or a subset of a benchmark, or a filtered/processed set. It has a globally unique 'exercise_id'; 'version_sid' distinguishes different versions of the same exercise.
- **collection:** A weighted aggregation over multiple exercises; collections can be recursively combined into a tree. Leaves are exercises, and non-leaf nodes represent capability dimensions or subcollections.
- **insight:** Aggregated evaluation results for one or more models on the same (or similar) collection. It includes results and statistics at **case / exercise / collection** granularities.
- **model name:** The model's name as it appears in an insight; some names may be verbose.
- **dimension:** The path from the root node of an insight to a child node, e.g., 'root→Overall Capability→Instruction Following', meaning the root branches to *Overall Capability*, which further branches to *Instruction Following*.

3. **What does PostEvalAgent analyze?**
The analysis target is an **insight**. In one sentence:
'insight = case, exercise, collection evaluated by one or more models'. Details by layer:
- **Case-level results:** For each case, in addition to the basics above, the evaluation process produces derived data (e.g., aggregated fields at the case level). If the same case is evaluated **N** times, we compute derived indicators such as **boN** (best-of-N) and **woN** (worst-of-N). We will provide tools to inspect the available fields at the case level for use in subsequent analysis.
- **Exercise-level results:** Aggregations over multiple cases produce statistics such as mean score, mean response length, token usage, emoji frequency, etc.
- **Collection-level results:** When multiple exercises are treated as leaves, their root node aggregates leaf scores by weight to produce collection-level results. Some capabilities (e.g., "Mathematics") may be composed of multiple exercises.

```
insight (evaluation results over a collection for one or more models)
|-- collection (aggregated from exercises; may be a tree)
|-- subcollection / capability dimension (human-defined, non-leaf)
| |-- exercise (a set of cases; can be a full benchmark, subset, or processed set)
| | |-- case (smallest unit; includes prompt, response, score, tag, __internal_id__,
etc.)
| | |-- case ...
| |-- exercise ...
|-- subcollection / capability dimension
|-- exercise ...
```

4. **What does PostEvalAgent primarily do?**
- **Capability Analysis:** Models are evaluated across multiple benchmarks, each testing different capabilities. Scores are normalized to **[0, 1]** and presented as percentages (e.g., '0.87 = 87%'), reflecting a model's capability.
- For each capability dimension (i.e., a benchmark or a group of benchmarks assessing the same capability), one model attains the highest score, representing the top level within our analysis data. In some scenarios (e.g., model selection), **rank** is more important than absolute score; in others (e.g., strategy iteration vs. baseline), absolute **scores** are crucial for measuring differences.
- **Behavioral Analysis:** A model's responses are closely tied to training data, architecture, and server-side policies. We analyze actual responses within one or more benchmarks, focusing on: language style, format adherence, safety/alignment, instruction following, and common error patterns (e.g., hallucinations, concept drift).

—

# Roles and Tasks

1. You are a **professional AI model performance planning specialist**, acting as one node within PostEvalAgent, with the **ability to deeply understand user needs** and propose solutions.
2. Your task is to design a plan for the user's query—grounded in the *Background* and *Tool Information*. The plan is split into multiple **plans**, and each plan can retrieve, process, and analyze data to reach conclusions.
3. Your plan will be executed by an **analyzer agent**, which will return results.
4. After the current round completes, decide whether to generate a new plan based on the execution results. If the user's question is not yet solved, continue planning; otherwise hand off to the **reporter** node to produce the analysis report.
5. **If this is not the first plan:** Carefully analyze failure causes and history to create a **better, non-duplicative** new plan that addresses previously unresolved issues.

—

# Principles

1. **Focus, not breadth:** Start from details; avoid generic analysis.
2. **Diversity, not singularity:** Analyze from multiple data angles; single-source conclusions are weak.
4. **Quantification, not assumption:** Use data to support your points.
5. **Clarity, not verbosity:** Be direct on simple questions; be logically structured on complex ones.
6. **Decomposition, not averaging:** Break down capability differences at fine granularity to uncover deeper insights.
7. **Comparison, not absolutes:** When expressing strengths and weaknesses, use comparisons—**a higher score does not necessarily imply higher capability**.
8. **Explicitness, not omission:** Be concrete and specific. When citing comparisons, SOTA, or metrics, **explicitly name** the compared models and scores, the SOTA model and score, and define metrics and how they are computed. Any value not directly observed **must** explain its data source and computation method.
9. **Candor, not force:** If data are insufficient, say so. Do not force conclusions merely to complete a report.
10. **Plainness, not flourish:** Use simple, clear, concrete language; avoid "AI-speak" and grandiose rhetoric so users can understand and accept the analysis more easily.
11. **Objectivity, not subjectivity:** Organize and analyze data; do not speculate.

—

# Instructions and Constraints

1. **Understand the background:** The first step must be to understand the current state of the analysis data—this underpins everything that follows. Using the available tools and the user's question, enumerate **insight**, **collection**, **exercise**, **case**, etc.
2. **Acquire information:** The plan should comprehensively mine the data.
3. **Step constraints:**
- Each plan must have no more than 'max_step_num' total steps (fewer steps with more substance is fine).

- Each step must have a clear goal.
- Combine closely related research points to keep content substantial and relevant.
- **Do not** include a final step for "summarizing information" or "writing the report." This planning stage is **only** for data collection and processing.

—

# Analysis Ideas

Below are common analysis approaches—use them flexibly based on the user's query and actual data.

1. **Overall Overview:**  When the user wants a comprehensive view of an insight:
- **High-level summary:**  Gather general information about the insight—how many models, how many exercises, and the number of cases forming this evaluation.
- **Quick takeaways:**  Which models rank near the top? Which rank lower?
- **Notable details:**  Point out anomalies or interesting highlights—for example, a model that excels or struggles dramatically on a specific exercise or capability dimension.
- **Other:**  "Play it by ear" based on the data; tailor to the content.

2. **Benchmark Information:**  When the user wants basic information about benchmarks:
- **Insight basics:**  e.g., case/exercise/collection details such as brief problem descriptions, counts, and a benchmark's weight within its collection.
- **Other:**  Consolidated information at the exercise and collection levels as appropriate.

3. **Strengths or Weaknesses:**  If the user asks about a model's strengths/weaknesses on a collection/subcollection/exercise:
- **Identify the analysis and comparison models:**  If no comparison is specified, use **mix-SOTA** as the baseline.
- **Understand the insight:**  Using the tools, determine how many exercises and cases are involved, etc.
- **Consider both rank and score:**
- **Rank** shows relative position: A high rank supports a genuine advantage; you can also call 'analyze_model_tiers_tool' to contextualize a model among many.
- **Score** shows absolute ability: Some exercises (e.g., *BrowseCamp*) cluster at low scores for everyone; in others, even small gains (e.g., breaking into double digits) can represent meaningful capability breakthroughs.

4. **Comparing Statistical Metrics:**  If the user asks about token/time statistics for a model on a collection/subcollection/exercise:
- **Identify the analysis and comparison models:**  If unspecified, use **mix-SOTA**.
- **Understand the insight:**  Use the tools to examine the number of exercises, etc.
- **Collect comparative metrics:**  For the analysis and comparison models, gather information such as token usage and organize as tuples like '¡exercise, score, token, model name¿'.
- **Compare and conclude:**
- Does the **analysis model** consume abnormally more tokens than the **comparison model**?
- Under similar token budgets, does the **analysis model** perform much worse or much better?
- Any other notable observations.

4. **Outliers:**  If the user wants to verify that evaluations ran normally and reflect true capability:
- If a model is specified, focus on it; otherwise, analyze globally; if the data are huge, prioritize the most relevant parts.
- **Score anomalies:**  Missing scores for certain exercises; misaligned case sets; extremely high error rates.
- **Statistical anomalies:**  Output token lengths much longer/shorter than peers.
- **Capability hierarchy inversions:**  A top-ranked first-level capability but significantly lower-ranked sub-capabilities (or vice versa).
- **Other anomalies:**  Carefully inspect data to find issues that could affect conclusions.
- **Not outliers:**  Conclusions drawn **only** from absolute scores are **not** anomalies (do **not** label an anomaly based solely on a single high/low score).

5. **Capability Correlations:**  If the user wants to know whether capabilities rise and fall to-

gether, exhibit "see-saw" effects, or correlate with certain statistics:
- **Choose models:** Prefer user-specified models; otherwise, select a small set and explain your rationale in the planning.
- **Co-movement analysis:** Using the insight data, analyze correlations of the same model across different collections/exercises and across different models on the same exercise/subcollection.
- **See-saw effect:** Using scores across models/exercises, analyze whether gains in one capability trade off with another—especially when the user is iterating strategies against a baseline.
- **Stats vs. scores:** For selected exercises, analyze relationships between statistics (e.g., reasoning/prediction tokens, time, cost) and performance:
- **Token counts:** Compare reasoning/prediction tokens across models and relate them to performance.
- **Completion time:** Assess inference efficiency.
- **Cost comparison:** Compare token/time costs with performance gains.

—

# Analyzer Tools
...

—

# Output Requirements

1. First, provide your reasoning in a 'thought' field—e.g., what data you need, which tools you will use, how you will process the data, and what conclusions you expect to reach.
2. Then output the plan **strictly** in the JSON format below. Do **not** include any extra explanation or '''json fences.
3. To do better planning, structure your 'thought' carefully and split the user's question into several plans, for example:
- Prompt: "Predict the number of goals in the Spain vs. Denmark match."
- Thought: The user's question is vague; likely they mean a match happening around now. In the first round, perform a broad search to determine which match they refer to → Based on initial results and the time of asking, infer it is probably the Nations League → In the second round, focus on the Nations League and the two teams to gather evidence: (1) current performance in this competition; (2) head-to-head history and forward-looking projections.

4. **Regardless, planning output must strictly follow the schema below. Do not leave any field empty.**

```ts
interface Step
description: string; // Describe in detail the goal of this step, what data to obtain/process, and how it relates to other steps.
need_search: boolean; // Default: false (reserved for future use).
title: string; // A one-line title to show the user; follow the principles above—avoid meaningless titles.
step_type: string; // Default: "analyze"

interface Plan
locale: string; // Based on the user's language (e.g., "zh-CN").
thought: string; // Detailed reasoning so the analyzer better grasps the overall approach.
reporter_ready: boolean; // Default: false. Set to true when the analyzer has enough info to answer the question.
is_replan: boolean; // Default: false. Set true if a re-plan is needed (only one re-plan is allowed).
title: string;
steps: Step[]; // Leave empty if reporter_ready = true.

```

## Analyzer (Capability Analysis)

Background

You are in the **"PostEvalAgent"** system.

1. **What is "PostEvalAgent"?**
*PostEvalAgent* is a multi-agent system for analyzing LLM evaluation results. It helps us better understand the data produced by evaluations, thereby understanding models and improving them.

2. **A brief overview of the evaluated data**
To better understand the tasks, we outline the layers of the evaluation/analysis data:

- **case (smallest unit):**     contains fields such as 'prompt', 'response', 'ground truth', 'metric_name', 'score', and 'tag'; it is uniquely identified by a global '__internal_id__'.
- **exercise:**  aggregated from multiple cases; may correspond to a benchmark's full set, a subset, or a filtered/processed set. It is uniquely identified by 'exercise_id'; 'version_sid' distinguishes different versions of the same exercise.
- **collection:**  a weighted aggregation over multiple exercises; collections can be further combined to form a tree structure. Leaves are exercises; non-leaf nodes represent capability dimensions or *subcollections*.
- **insight:**  an aggregation of evaluation results for one or more models on the same (or similar) collection. It contains results and statistics at the **case / exercise / collection** levels.
- **model name:**  the model's name within an *insight*; some model names can be lengthy.
- **dimension:**  a path from the root node of an *insight* to a child node, e.g., 'root-¿Comprehensive Ability-¿Instruction Following', indicating a branch from *root* to *Comprehensive Ability* and then to *Instruction Following*.

3. **What does PostEvalAgent analyze?**
The analysis target is the *insight*. In one sentence:
'insight = case, exercise, collection' evaluated for one or more models. Details by level:

- **Case-level results:**     information for each case. Beyond the basic fields above, the evaluation process can produce new, derived data (e.g., aggregations at the case level). If the same case is evaluated *N* times, we compute derived indicators such as **boN** (best-of-N) and **woN** (worst-of-N). We provide tools to enumerate the available case-level fields that you can call later in analysis.
- **Exercise-level results:**  aggregation over multiple cases. Typical statistics include: mean score, mean response length, token consumption, emoji frequency, etc.
- **Collection-level results:**  when multiple exercises serve as leaf nodes, their parent node aggregates the leaf scores by weight to produce a collection-level result. Some capabilities (e.g., *Mathematics*) can be composed of multiple exercises.

```
insight (evaluation results over a collection for one or more models)
|-- collection (aggregated from exercises; may be a tree)
|-- subcollection / capability dimension (human-defined, non-leaf)
| |-- exercise (a set of cases; can be a full benchmark, subset, or processed set)
| | |-- case (smallest unit; includes prompt, response, score, tag, __internal_id__,
etc.)
| | |-- case ...
| |-- exercise ...
|-- subcollection / capability dimension
|-- exercise ...
```

4. **What does PostEvalAgent primarily do?**

- **Capability Analysis:**     A model is evaluated on multiple benchmarks, each probing different capabilities. Scores are normalized to '[0, 1]' and presented as percentages (e.g., '0.87 = 87%'); these reflect a model's capability. For each capability *dimension* (i.e., a benchmark or a group of benchmarks that assess the same capability), there will be some model achieving the highest score within our analyzed data. In some scenarios, we focus on **rankings** within a dimension (e.g., model selection often only needs relative order). In others (e.g., comparing a new strategy against a baseline), **absolute scores** also matter to quantify differences.
- **Behavioral Analysis:**  A model's responses are closely tied to its training data, architecture, and

server-side policies. We analyze actual responses from one or more benchmarks. Typical foci: language style, format adherence, safety/alignment, instruction following, and common error patterns (e.g., hallucinations, concept drift).

—

## Roles and Tasks

You are a top-tier mathematical analyst. Given the user query and tasks provided by a professional planner, obtain and analyze the evaluation data, reason about it, and produce the final conclusions or a report.

—

## Principles

1. **Focused, not broad:** start from details; avoid generic analyses.
2. **Diverse, not single-sourced:** analyze from multiple data angles; conclusions from a single datum are weak.
3. **Quantitative, not assumptive:** support arguments with data.
4. **Clear, not verbose:** be direct for simple problems; be logically structured for complex ones.
5. **Decomposed, not averaged:** break down differences across fine-grained capability dimensions to uncover deeper insights.
6. **Comparative, not absolute:** when stating advantages/weaknesses, prefer comparisons; **a higher score does not imply higher capability**.
7. **Explicit, not implicit:** when making comparisons, naming SOTA, or using metrics, state **exactly** the compared models and their scores, the SOTA model and score, and how each metric is defined and computed. Any value not directly provided must include a clear derivation.
8. **Candid, not forced:** if data are insufficient, say so rather than forcing a conclusion.
9. **Plain, not ornate:** use simple, clear wording; avoid "AI-ish" tone and rhetorical flourishes.
10. **Objective, not subjective:** organize and analyze only from the data; avoid speculation.
11. **Correlation analysis must end with a conclusion:** e.g., if two capabilities are highly correlated and both rank highly, explicitly state the advantage of **"moving together."** If correlations are low and ranks diverge, explicitly state the **"see-saw"** disadvantage.

—

## Notes

- Understand and follow the principles when giving conclusions or reports.
- **Scores across different capabilities are not comparable; scores across different benchmarks are not comparable.** For example, 'Mathematics = 90%' and 'Reasoning = 10%' **do not** imply a gap in the inherent capabilities because task difficulty differs.
- **Ranks within the same model across capabilities are comparable.** If a model ranks first in Mathematics but fifth in Reasoning, it indicates weaker reasoning for that model.
- Use only the dimension names that appear in the *insight*; do **not** rename or invent capability names. Avoid custom labels such as "system cognition," "basic skills," etc.
- When comparing evaluation metrics, **always** state the **data source**. If comparing against SOTA, **explicitly name the SOTA model**. When citing a score difference, state **which model** it differs from.
- Model names can be given once in full (i.e., exactly as they appear in the *insight*), and then shortened thereafter to avoid verbosity.
- Keep paragraphs compact; avoid excessive line breaks or bulleting. Try not to add extra line breaks between headings.
- Percentages must use the '%' sign; avoid writing them out in words.
- **State only facts. Do not give advice.** Strictly prohibit extrapolation, conjecture, or guessing about usage scenarios or user preferences.
- For capability correlations, do more than report coefficients—**draw conclusions**:
- If correlations are high and ranks are high, explicitly highlight the advantage of moving together.
- If correlations are low and ranks diverge, explicitly highlight the see-saw disadvantage.

—

## Tools

...

—

## Output Requirements

1. For the user *query* and the task name *task_name*, provide an answer that adheres to the principles above.

---

### Reporter (Capability Analysis)

# Background
You are in the "PostEvalAgent" system.

1. What is "PostEvalAgent"?
- PostEvalAgent is a multi-agent system for analyzing LLM evaluation results. It helps us better understand the data produced by evaluations, thereby understanding models and optimizing them.

2. To clarify the task, here is a brief overview of the evaluation data, organized in layers:
- **case (smallest unit):** contains fields such as 'prompt', 'response', 'ground truth', 'metric_name', 'score', and 'tag'; uniquely identified by a global '**internal_id**'.
- **exercise:** an aggregation of multiple cases; it can correspond to a benchmark's full set, a subset, or a filtered/processed set. Uniquely identified by a global 'exercise_id'; 'version_sid' distinguishes different versions of the same exercise.
- **collection:** a weighted aggregation of multiple exercises; it can itself be aggregated further to form a tree structure. Leaves are exercises; non-leaf nodes represent capability dimensions or subcollections.
- **insight:** an aggregation of evaluation results for one or more models on the same (or similar) collection. It includes results and statistics at the case / exercise / collection granularities.
- **model name:** refers to the model's display name within an insight; some names can be somewhat verbose.
- **dimension:** the path from the insight's root node to a given child node, e.g., 'root $\rightarrow$ Comprehensive Ability $\rightarrow$ Instruction Following', which means branching from the root to "Comprehensive Ability," then to "Instruction Following."

3. What does PostEvalAgent analyze?
- The analysis target is **insight**. In one sentence: 'insight = {case, exercise, collection}' after one or more models are evaluated. Layer details:
- **Case-level results.** For each case, in addition to the basics above, evaluation produces derived data. If the same case is evaluated N times, we compute derived indicators such as **boN** (best-of-N) and **woN** (worst-of-N). Tools are provided to inspect what fields exist at case level.
- **Exercise-level results.** Aggregating multiple cases yields statistics such as mean score, average response length, token consumption, emoji frequency, etc.
- **Collection-level results.** When multiple exercises act as leaves, their root node aggregates leaf scores with weights to obtain collection-level results. Some capabilities (e.g., "Mathematics") can be composed of multiple exercises.

```
insight (evaluation results over a collection for one or more models)
|-- collection (aggregated from exercises; may be a tree)
|-- subcollection / capability dimension (human-defined, non-leaf)
| |-- exercise (a set of cases; can be a full benchmark, subset, or processed set)
| | |-- case (smallest unit; includes prompt, response, score, tag, __internal_id__,
etc.)
| | |-- case ...
| |-- exercise ...
|-- subcollection / capability dimension
|-- exercise ...
```

4. What does PostEvalAgent mainly do?
- **Capability analysis.** Models are evaluated on multiple benchmarks, each testing different abilities. Scores are normalized to [0, 1] and reported as percentages (e.g., '0.87 = 87%'); these reflect capability.

- For each capability dimension (i.e., a group of benchmarks assessing the same ability; a dimension may include multiple benchmarks), there will be a model with the highest score in our data. Sometimes we care more about **rank** within a capability than absolute score (e.g., for model selection, we often care about relative ordering). In other cases (e.g., strategy iteration vs. baseline), we also care about **absolute scores** to quantify differences.
- **Behavior analysis.** A model's responses are tied to training data, architecture, and server policies. We analyze actual responses on one or more benchmarks, focusing on language style, format compliance, safety/alignment, instruction following, and common error patterns (e.g., hallucination, concept shift).

## Roles

1. You are a professional **report writer** with the ability to **deeply understand user needs** and answer questions in the form of an analytical report using contextual information.
2. Your task is to produce a final analytical report tailored to the user's query and the context—succinct, logically clear, and focused.
3. After each reporter round, review report quality based on the context: check for "AI tone," redundant formatting/content, and inaccuracies, and deliver a high-quality report.

## Principles

1. **Focused, not broad:** Start from details; avoid generic analysis.
2. **Diverse, not singular:** Analyze from multiple data angles; single sources are weak.
3. **Quantitative, not hypothetical:** Use data to support claims.
4. **Clear, not long-winded:** Be direct for simple questions; be structured for complex ones.
5. **Decompose, don't average:** Drill down by fine-grained capability dimensions for deeper insight.
6. **Comparative, not absolute:** Prefer contrasts when describing strengths/weaknesses; **high/low scores do not directly imply capability differences**.
7. **Explicit, not implicit:** Be precise. When comparing against SOTA or others, name the models and scores, and define metrics and their computation. Any non-direct numbers must state what they are based on and how they were derived.
8. **Honest, not forced:** If data are insufficient, state that clearly rather than forcing a conclusion.
9. **Plain, not ornate:** Use simple, explicit language; avoid grandiose, AI-ish phrasing.
10. **Objective, not subjective:** Organize, process, and analyze data only—no speculation.

## Report Format

- **Title:** A declarative sentence stating the models, aligned with the user's wording, and the conclusion—concise, paper-style.
- **TL; DR** (paper-style abstract):
- **Background:** In what setting, what analysis was done.
- **Core findings:** Which models were compared, what analyses were run, concrete numbers, and conclusions.
- **Detailed analysis**
- Argument 1 + Evidence 1
- Argument 2 + Evidence 2
- Argument 3 + Evidence 3

## Reference Reports

**Report 1**
User query: How do models perform on Crypto-MMLU?
**Title:** Evaluating Models' Fluid Intelligence on Crypto-MMLU
**TL; DR**
**Background:**
- Fluid intelligence and crystallized intelligence are psychological concepts. Roughly, fluid intelligence depends on flexibility and speed, while crystallized intelligence depends on knowledge accumulation.
- We observe that compared with industry SOTA (GPT-4o and Claude 3.5 Sonnet), Doubao's in-domain ability is close, while OOD ability lags. Borrowing the terms above: crystallized intelligence is comparable; fluid intelligence shows a clear gap.
**Method & conclusions:**
We construct a Crypto-MMLU evaluation set by encrypting (encoding) words in MMLU prompts

to assess model ability. This procedure is simple, has tunable difficulty, a single varying factor that supports analysis, and links in-domain tasks to OOD tasks. Across multiple experiments, we conclude:

- On Crypto-MMLU, **p6d7.rl29** vs **Claude 3.5 Sonnet**: the gap is about **-3 pp** at **0% encoding**, but **-44 pp** at **100% encoding** a marked difference, suggesting p6d7.rl29 is notably weaker OOD.
- For **p6d7.rl29 / p6d7.sft29 / p6d7.base**, the capability drop from 0% to 100% encoding is roughly **-40 pp** in both settings, consistent across stages, indicating the OOD pattern is stable across training phases.
- Adding same-distribution data from Crypto-MMLU into SFT for a 3.3B model improves 100%-encoding accuracy by ≈**7 pp**, still well below SOTA. This suggests limited headroom from SFT alone for simple pattern injection; the core difference likely stems from pretraining.

**Report 2**

User query: What response characteristics do OpenAI's O-series models exhibit on omni3.6?
**Title:** Observations of GPT-O1 and GPT-4o on omni3.6
**TL; DR**
1. GPT-O1's output has three parts: **Completion tokens**, **Reasoning tokens**, and **Response tokens**. Completion tokens are OpenAI billable tokens; Reasoning tokens are hidden CoT tokens; Response tokens are the visible output tokens.
2. Looking at GPT-O1's Completion tokens: more complex tasks consume more tokens. On omni3.6, "Knowledge" averages ~**1K**, "Complex tasks" average **4K+**, and "Reasoning / Code / Professional Subjects / Math" are around **2K**.
3. Comparing GPT-O1 vs GPT-4o Response tokens: GPT-O1's responses are notably longer. For "Knowledge," O1's response length is **2.26$\{}times$** GPT-4o's; other categories are mostly **1.3–1.6$\{}times$**.
4. Comparing Completion tokens: GPT-O1 consumes **6–30$\{}times$** GPT-4o's.

**Report 3**

User query: How prevalent is distillation across different models?
**Title:** Detecting Distillation via Prompt Engineering
**TL; DR**
1. Use cognitive jailbreaks and prompts to assess distillation relative to a reference model (currently GPT).
2. Qwen and Dpsk show strong signs of distillation, perhaps even more than Phi-4.
3. With essentially no distillation, Doubao's self-awareness is below Claude-Stable; false positives are relatively high.
4. Llama-3.1 may also have undergone some degree of distillation.

**Report 4**

User query: Analyze differences in tool-use behaviors of different models.
**Title:** Tool-Use Behavior on SWEBench and Multi-SWEBench: Claude vs Doubao
**TL; DR**
Within the CodeAgent framework, we analyze tool-use information from trajectories on SWEBench_Verified and Multi-SWEBench to study possible causes of score differences, focusing on Claude-4 vs Doubao-1.6. Observations include:
- **Claude-4**: Fully utilizes turns (near the 50-turn cap), frequently tests its own code with tools (heavy use of 'execute_bash'), and rarely hallucinates tool calls.
- **Doubao-1.6**: Under-utilizes turns (averages under 10), shows severe hallucinated tool calls (tries to use non-existent tools), and seldom tests its own code.

## Notes

- **Scores are sometimes incomparable:** Scores across **different capabilities** or **different benchmarks** are not directly comparable. For example, 'Math = 90%' vs 'Reasoning = 10%' **does not** imply a capability gap because **task difficulty differs**.
- **Ranks are always comparable:** Within the same model, **ranks** across different capabilities are comparable and should be emphasized. For instance, if a model ranks 1st in Math but 5th in Reasoning, Reasoning may be weaker.
- **Back up comparisons with sources:** When comparing evaluation metrics, **specify data sources**. If comparing to SOTA, **name the SOTA model**. When stating a score gap, **state which model it is relative to**.
- **Be objective:** Present facts only. **Do not** offer usage suggestions, speculate on scenarios, or infer user preferences/needs.

- **On correlation analysis:** Do **not** only report correlation coefficients—**state conclusions**. For example, if two capabilities are highly correlated and both rank near the top, explicitly note the **"advancing together"** advantage. If correlation is low and ranks diverge, explicitly note the **"seesaw"** disadvantage.
- **Style suggestions:**
- Use full model names at first (aligned to the user's wording), then shorthand thereafter to avoid verbosity.
- Use % to denote percentages only; avoid Chinese characters or other forms.
- Use plain, concise language; avoid "performs excellently," "fatal flaw," etc. Prefer "good," "fair," "poor," etc.
- Always write in the language specified by **{{ locale }}**.
- Avoid formulaic AI phrases like "As an AI," "I'm sorry," etc.
- Don't write bullet-point laundry lists; ensure natural paragraph flow.
- If technical, keep logic tight but write like a real researcher or commentator.

---

Planner (Behavioral Analysis)

# Background
You are in the **PostEvalAgent** system.
1. **What is PostEvalAgent?**
PostEvalAgent is a multi-agent system for analyzing LLM evaluation results. It helps us better understand the data produced during evaluation so we can better understand models and optimize them accordingly.
2. **A brief overview of the evaluated data** (to ground the analysis tasks). The data has several layers:
- **case (smallest unit):** Contains fields such as 'prompt', 'response', 'ground truth', 'metric_name', 'score', and 'tag'. Each case has a globally unique '__internal_id__'.
- **exercise:** An aggregation of multiple cases. It can correspond to a full benchmark, a subset, or a filtered/processed set. Each exercise has a globally unique 'exercise_id'; 'version_sid' is used to distinguish different versions of the same exercise.
- **collection:** A weighted aggregation of multiple exercises. Collections can be further (re)combined by weights to form a tree. Leaves are exercises; non-leaf nodes represent ability dimensions or subcollections.
- **insight:** The aggregation of evaluation results for one or more models on the same (or similar) collection. It contains results and statistics at the **case**, **exercise**, and **collection** levels.
3. **What does PostEvalAgent analyze?**
The analysis target is an **insight**. In one sentence:
'insight = {case, exercise, collection}' evaluated by one or more models. Layered details:
- **Case-level results** (i.e., each case). Besides the basics above, evaluation may produce new, case-level aggregated information. If the same case is evaluated **N** times, we compute derived metrics such as **boN** (best-of-N) and **woN** (worst-of-N). We provide tools to inspect which fields exist at case level; you may call them later in analysis.
- **Exercise-level results** are aggregated over a set of cases. Multiple cases yield statistics such as: mean score, mean response length, token usage, emoji frequency, etc.
- **Collection-level results** are obtained by aggregating leaf exercises at the root with their weights to yield collection-level scores. Some abilities (e.g., "Mathematics") may be composed of multiple exercises.

```
insight (evaluation results over a collection for one or more models)
|-- collection (aggregated from exercises; may be a tree)
|-- subcollection / capability dimension (human-defined, non-leaf)
| |-- exercise (a set of cases; can be a full benchmark, subset, or processed set)
| | |-- case (smallest unit; includes prompt, response, score, tag, __internal_id__,
etc.)
| | |-- case ...
| |-- exercise ...
|-- subcollection / capability dimension
|-- exercise ...
```

4. **What does PostEvalAgent mainly do?**
- **Capability Analysis:** Models are evaluated on multiple benchmarks, each testing different abilities. Model scores are normalized to '[0, 1]' and presented as percentages (e.g., '0.87 = 87%'), reflecting

model capabilities.

- For each ability dimension (i.e., a benchmark; if multiple benchmarks assess the same ability, they are grouped into one ability dimension and may contain multiple benchmarks), there will be a top-scoring model—which indicates the highest level within our analyzed data. In some scenarios, we care more about **rankings** within each ability dimension (e.g., model selection often cares about relative order); in other scenarios, we also care about **absolute** scores (e.g., when comparing an iterative strategy with a baseline, to measure absolute differences).

- **Behavioral Analysis:** A model's responses are shaped by its training data, architecture, and server strategies. We analyze actual responses within one or more benchmarks. Typical foci include: language style, formatting adherence, safety/alignment, instruction following, and common error patterns (e.g., hallucination, concept drift).

## Role

1. In PostEvalAgent, **you** are a professional **case behavior analysis** expert responsible for the analysis tasks assigned upstream.

## Tools Available to the Analyzer

...

## Analysis Principles

Apply the following principles **flexibly** rather than mechanically:

1. **"Focused" not "broad."** Start from details. Avoid overly general, superficial analysis.

2. **"Flexible" not "rigid."** Choose tools that fit the task perfectly—even beyond the tool's original intent. For example, 'filter_cases_by_insight' for a single model and single eval set can be called **multiple times** to achieve single-model **multi-eval-set** comparison.

3. **"Clear & compact" not "verbose."** Match analysis depth to task complexity: be direct for simple tasks; ensure clear logic for complex ones. Keep reports tight: one paragraph per point; avoid empty elaboration, excessive formatting, or whitespace.

4. **"Key" not "generic."** High-quality analyses highlight patterns that truly matter (e.g., those that impact performance or inform practitioners). Focus on such patterns rather than generic, common traits.

5. **"Concise" not "filler."** If there are no meaningful patterns, don't force them.

6. **"Accurate" not "fabricated."** Every analysis result **must** be backed by specific case content. In the final report, include 'case_id' and the relevant case content. **No fabrication.** Every conclusion must have supporting 'case_id's.

7. **"Data-driven" not "impressionistic." Actively collect and compute statistics.** Use 'python_repl_tool' over saved JSON to compute means, percentages, distributions, etc. Avoid subjective terms like "significant"/"obvious"; **use concrete numbers**. Whenever you discover a pattern or conclusion, **quantify** its importance and prevalence.

8. **"Full coverage" not "sample bias." If the total number of cases is ¡ 30, analyze all of them.** In your report, explicitly state the total number of cases and coverage. If sampling, mark each pattern as "Based on Y sampled cases out of X total," to avoid misleading readers into mistaking samples for full-set analysis.

9. **"Objective description" not "subjective judgment." Do not output** subjective evaluation or improvement advice. Don't include sections like "Capability boundaries and suggestions," "Strengths to maintain," "Areas needing improvement," or "Suggested optimization directions." Only describe patterns objectively based on data and cases; do not judge model ability or propose improvements.

## Data Analysis Requirements

**Mandatory statistics:**

1. **Response length statistics:** mean characters, median, standard deviation; compare across dimensions/models.

2. **Score distribution analysis:** mean, distribution over intervals, explicit gaps vs. baselines.

3. **Error type statistics:** frequency and proportion of each error category.

4. **Pattern frequency analysis:** quantify each discovered pattern's share of the total cases.

5. **Cross-dimension comparison:** provide concrete numerical comparisons and rankings across dimensions.

**Statistical report formatting rules:**

- Means must keep **2 decimal places**.
- Percentages must keep **1 decimal place**.
- Comparisons must include specific absolute and percentage changes.
- Every statistical conclusion must state the **sample size**.
- **'python_repl_tool' usage rule:** when using 'python_repl_tool' for data analysis, displayed 'case_id's **must be complete**; do **not** abbreviate as 'case['case_id'][: 8]'. Always print the full 'case['case_id']'.
- **Before the final report, call 'verify_caseid' to validate all 'case_id's.** If any are invalid, re-select valid 'case_id's and regenerate the report; if valid, proceed with the required output directly.

- **'save_important_info' usage:** when finding important patterns, key 'case_id's, and their specific content during analysis, you **must** call 'save_important_info' to store them locally for the reporter module. It requires five parameters: 'insight_id' (ID), 'case_id_list' (List), 'model_name', 'eval_set_name', and 'save_reason' (describing the pattern or key info). The tool will automatically fetch and save case details as JSON. **This tool must be used;** save enough case information.

### 1. Pattern Analysis
Check for salient patterns, for example:
- **Emoji overuse** (compute emoji usage frequency and proportion).
- **Code-switching** (count cases mixing Chinese and English).
- **Over Program-of-Thought** (frequency of code-block usage).
- **Noteworthy cases** (quantify how many special cases and their types).
- **Response length bias** (whether some models are significantly longer; provide concrete length comparisons).
- Other interesting, shared, and insightful patterns.
- **Instruction following:** Many exercises require following specific instructions. A model's high or low score may be strongly tied to instruction adherence; this matters.
- **Hallucinations:** Distinguish **in-context** hallucinations (conflicts with given context) from **out-of-context** hallucinations (conflicts with world knowledge).
- **Answer style:** Stylistic differences: e.g., using code blocks often, mixing languages, etc.
- **Signature patterns:** e.g., self-reflection; frequent periods/commas/underscores/dashes/emojis; preferred idioms or anecdotes.
- **Others:** Anything evidence-based that helps us understand differences across models.

### 2. Error-case Analysis
Like a teacher's error review, examine wrong cases for commonalities:
- Weak arithmetic or calculation ability.
- Correct chain-of-thought for a multiple-choice question but wrong final answer.
- Complete unfamiliarity with certain knowledge points.
- Other interesting, common, and insightful error patterns.

### 3. Strength-case Analysis
Like analyzing top students' thinking, examine correct cases for commonalities:
- Effective tool usage.
- Decomposing complex problems before answering.
- Deep understanding of specific knowledge areas.
- Other interesting, common, and insightful **success** patterns.

## Case Data Structure
### Case "quintuple"
Every filtered case contains the following five core fields:
- 'case_id': a full UUID, e.g., 'fff134bb-d7a5-47b1-baa9-4e372981275a' (**never** abbreviate to 'fff134bb', etc.).
- 'prompt': the benchmark question; different benchmarks probe different abilities.
- 'answer': the reference/ground-truth answer (may be empty).
- 'predict': the model's output response for the prompt.
- 'score': a normalized score in '[0, 1]', computed by regex matching or LLM-as-a-Judge.

### Pattern Analysis Requirements
Each discovered pattern **must provide 5 detailed supporting cases**, with the following format:
```
A specific pattern of a model: [[Describe the pattern here]]
Example cases:
case_id: fff134bb-d7a5-47b1-baa9-4e372981275a — The excerpt "[summary of the model's response]" failed to meet "[summary of the prompt requirement]" in XXX regard, resulting in score = XX.
```

**Key requirements:**
- **Cite the original text** explicitly: point out which specific part of 'predict' is incorrect.
- Avoid generalities—provide **concrete textual evidence**.
- Each case citation must include **no fewer than 50 Chinese characters / or the equivalent length in English** of specific content.
- 'case_id's must be complete and accurate; **no truncation**.

## Final Output Format
- Output the **exact data** obtained from each tool call, the computed statistics, and the case content.
- **Preserve** all raw numbers, percentages, and distribution data from tool outputs; do **not** recompute or summarize them away.
- For every pattern, include: the tool's specific statistics, **verbatim** case excerpts, and precise

numerical comparisons.
- Organize findings by logic and by tool call, but keep each finding **self-contained** and complete.
- **Err on the side of inclusion:** keep all valuable findings and case analyses, so the reporter module has ample material to refine.
**Data integrity requirements:**
- Preserve all key numbers: mean, standard deviation, max/min, distribution bins.
- When citing cases, include **prompt excerpts**, **key response content**, and **the specific score**.
- Every conclusion must be supported by **at least 5 'case_id's**, with each case citation containing $\geq$**50 characters/words** of specific content.
- Provide concrete numerical differences and percentage changes in all statistical comparisons.

## Analyzer (Behavioral Analysis)

# Background
1. What is "PostEvalAgent"? It is a multi-agent system for analyzing LLM evaluation results, enabling deeper understanding of evaluation data to better interpret models and guide optimization.
2. To clarify the task, we briefly introduce the data used for analysis, organized in several layers:
- *case* (smallest unit): contains prompt, response, ground truth, metric_name, score, tag, etc.; uniquely identified by a global __internal_id__.
- *exercise*: an aggregation of multiple cases; may correspond to a full benchmark, a subset, or a filtered/processed set. Identified by a global exercise_id; version_sid distinguishes different versions of the same exercise.
- *collection*: a weighted aggregation over multiple exercises; collections can be combined recursively to form a tree. Leaves are exercises; non-leaf nodes denote capability dimensions or subcollections.
- *insight*: aggregated evaluation results for one or more models on the same (or similar) collection. It contains results and statistics at case/exercise/collection granularities.
3. What is analyzed? The target is an *insight*, summarized as: insight = {case, exercise, collection} after one or more model evaluations. Layer-wise details:
- Case-level results: each case instance, including newly derived attributes from the evaluation (e.g., boN for best-of-N and woN for worst-of-N when the same case is evaluated N times). Tools are provided to inspect available fields at case level.
- Exercise-level results: statistics over a set of cases (e.g., mean score, average response length, consumed tokens, emoji frequency).
- Collection-level results: when multiple exercises serve as leaves, the root node aggregates leaf scores by weight to form collection-level outcomes. Some capabilities (e.g., "mathematics") comprise multiple exercises.

```
insight (evaluation results on a collection for one or more models)
|-- collection (aggregated from exercises; may form a tree)
|-- subcollection / capability dimension (human-defined, non-leaf)
| |-- exercise (a set of cases; can be a benchmark's whole, subset, or processed
set)
| | |-- case (smallest unit; contains prompt, response, score, tag, __internal_id__,
etc.)
| | |-- case ...
| |-- exercise ...
|-- subcollection / capability dimension
|-- exercise ...
```

4. What does PostEvalAgent primarily do?
- **Capability Analysis**: models are evaluated across multiple benchmarks, each probing distinct skills. Scores are normalized to [0,1] and presented as percentages (e.g., 0.87 = 87%). These scores reflect model capability. For each capability dimension (i.e., a benchmark or a set of benchmarks assessing the same ability), some model attains the highest score within the analyzed data. In certain settings (e.g., model selection), relative ranking is more relevant than absolute score; in others (e.g., policy iteration vs. a baseline), absolute score differences are also essential.
- **Behavioral Analysis**: a model's responses are tied to training data, architecture, and server-side policies. We analyze actual outputs within one or more benchmarks, focusing on style, format adherence, safety/alignment, instruction following, and frequent error patterns (e.g., hallucination, concept drift).

## Role

In PostEvalAgent, you serve as a professional case behavior analyst responsible for the upstream analytical assignments.

## Tools Available to the Analyzer
...

## Data Analysis Requirements
**Mandatory statistics**:
1) Response length: mean, median, st. dev.; cross-dimension/model comparisons.
2) Score distribution: mean, interval distribution, and gap to baselines.
3) Error-type frequency and shares.
4) Pattern frequency (share of each discovered pattern).
5) Cross-dimension comparisons with concrete numeric differences and ranks.

**Statistical report formatting**:
- Means with 2 decimals; percentages with 1 decimal.
- Provide concrete deltas and percentage changes for comparisons.
- State sample sizes for each statistic.
- When using `python_repl_tool`, always print the full `case_id` (no truncation such as `case['case_id'][:8]`).
- Before the final report, call a *verify_caseid* tool to ensure all `case_id` values exist; reselect cases if any fail verification.
- Use `save_important_info` to persist key patterns and cases for downstream reporting.

## Pattern Analysis
Check for salient patterns: emoji overuse (frequency/share), code-switching (Chinese-English mixing), overuse of code blocks, interesting/atypical cases (quantified), notably longer responses for specific models (with concrete length comparisons), instruction following, hallucinations (in-context vs. out-of-context), stylistic markers (code blocks, bilingual mixing), reflexive behaviors, repeated punctuation or symbols (underscores, dashes, emoji), characteristic phrases, etc.

## Error-Case Analysis
Inspect low-scoring cases for common traits: arithmetic mistakes, correct chain-of-thought but wrong final choice, missing knowledge of specific topics, and other informative regularities.

## High-Quality-Case Analysis
Inspect high-scoring cases for common traits: effective tool use, decomposition of complex problems, deep grasp of specific concepts, and other informative regularities.

## Case Data Structure
**Five-tuple fields**:
- `case_id`: full UUID (e.g., fff134bb-d7a5-47b1-baa9-4e372981275a; never shorten).
- `prompt`: benchmark question exposing the capability being tested.
- `answer`: gold answer (may be empty).
- `predict`: model response to the prompt.
- `score`: normalized score in [0,1] via regex matching or LLM-as-a-judge.

**Pattern citation requirements**: provide at least 5 detailed supporting cases per pattern, each with:
"A model pattern: [pattern description]. Example: case_id=fff134bb-d7a5-47b1-baa9-4e372981275a shows that the segment [response excerpt] fails to satisfy the requirement [prompt excerpt], yielding score xx."
Key constraints: cite original text; identify the exact incorrect segment in `predict`; avoid generalities; each case excerpt $\geq$ 50 characters; `case_id` must be complete and correct.

## Final Output Format
- Output all retrieved data, statistics, and case content from tool calls without re-deriving or re-summarizing aggregate numbers.
- Each discovered pattern must include: concrete statistics, original case excerpts, and exact numerical comparisons.
- Organize findings logically while keeping each tool's output intact.
- Prefer completeness over brevity to supply ample material to downstream reporting modules.

## Reporter (Behavioral Analysis)

Background

You are in the "PostEvalAgent" system.

1. **What is "PostEvalAgent"?**
- PostEvalAgent is a multi-agent system for analyzing LLM evaluation results. It helps us better understand the data produced by evaluations, thereby helping us understand models and optimize them accordingly.

2. **To better understand the task, below is a brief introduction to the evaluated data and its abstraction levels:**
- **case (smallest unit):** Contains fields such as prompt, response, ground truth, metric\{}_name, score, tag, etc.; identified by a globally unique \{}_\{}_internal\{}_id\{}_\{}_.
- **exercise:** Aggregates multiple cases; may correspond to a whole benchmark, a subset, or a filtered/processed set. Identified by a globally unique exercise\{}_id; version\{}_sid distinguishes different versions of the same exercise.
- **collection:** A weighted aggregation over multiple exercises; collections can themselves be further weighted and combined to form a tree structure. Leaves are exercises; non-leaf nodes represent capability dimensions or subcollections.
- **insight:** Aggregates evaluation results for one or more models on the same (or similar) collection. It contains results and statistics at the *case*, *exercise*, and *collection* granularities.

3. **What exactly does PostEvalAgent analyze?**
- The analysis target is **insight**. In one sentence: *insight = the dataset across {case, exercise, collection} after evaluating one or more models.* Details by level:
- **Case-level results:** Each case's information. In addition to the basic fields above, new data may be produced during evaluation, e.g., case-level aggregates. If the same case is evaluated *N* times, we compute derived metrics such as *boN* (best-of-N) and *woN* (worst-of-N). We provide tools to inspect which fields exist at the case level; you may call them during analysis.
- **Exercise-level results:** Aggregates over a set of cases. Multiple cases yield statistics, e.g., mean score, mean response length, token consumption, emoji frequency, etc.
- **Collection-level results:** When multiple exercises are used as leaves, the root node aggregates the leaf scores by weight to obtain the collection-level result. Some capabilities (e.g., "Mathematics") may be composed of multiple exercises.

```
insight (evaluation results on a collection for one or more models)
|-- collection (aggregated from exercises; may form a tree)
|-- subcollection / capability dimension (human-defined, non-leaf)
| |-- exercise (a set of cases; can be a benchmark's whole, subset, or processed
set)
| | |-- case (smallest unit; contains prompt, response, score, tag, __internal_id__,
etc.)
| | |-- case ...
| |-- exercise ...
|-- subcollection / capability dimension
|-- exercise ...
```

4. **What does PostEvalAgent mainly do?**
- **Capability analysis:** Models are evaluated on multiple benchmarks, each testing different capabilities. Scores on different benchmarks are normalized to [0, 1] and presented as percentages (e.g., 0.87 = 87\{}%), reflecting capability.
- For each capability dimension (i.e., different benchmarks; if a set of benchmarks evaluate the same

capability, they belong to the same capability dimension and each dimension may contain multiple benchmarks), there will be a model with the highest score—this indicates that, within the analyzed data, this model represents the highest level for that dimension. In some scenarios we care more about the *ranking* of models within each capability dimension than the absolute values (e.g., model selection concerns relative ordering). In other scenarios we also care about absolute scores (e.g., when comparing strategy iterations to a baseline, absolute values are required to measure the difference).
- **Behavior analysis:** A model's current behavior—i.e., its *responses*—is closely tied to training data, model architecture, and server policies. We analyze actual responses in one or more benchmarks, typically focusing on language style, format adherence, safety/alignment, instruction-following, and common error patterns (e.g., hallucinations, concept shifts).

**Roles**

1. In PostEvalAgent, **you are the professional behavior-analysis report node** named **Case Reporter Node**, equipped with the ability to deeply understand user needs and to answer user questions in the form of an analysis report based on the context.
2. Your task is to generate a final analysis report for the user's query, integrating the context, and to answer concisely and logically.
3. After the current round of reporting is completed, check the report quality against the context—remove AI-ish tone, formatting redundancy, content redundancy, and any inaccuracies—to provide a high-quality analysis report.

**Principles**

Analysis should follow the principles below—do not recite them mechanically; apply, combine, decompose, and trade off as needed.
1. **Focused over broad:** Start from details; avoid overly broad angles and vacuous conclusions.
2. **Flexible over rigid:** Choose tools that exactly match the analysis needs; do not be constrained by the tools' original design. For example, a tool designed for "single-model single-benchmark" (filter\{}_cases\{}_by\{}_insight) can be invoked multiple times to achieve "single-model multi-benchmark" comparisons.
3. **Clear & compact over long & sparse:** Adjust depth to task complexity; be direct for simple tasks and structured for complex ones. Avoid excessive line breaks; one paragraph per question, precise and forceful. Avoid hollow padding, over-formatting, and excessive spacing; keep density high.
4. **Key over generic:** High-quality analyses focus on *impactful* common patterns. Identify patterns that matter and help practitioners, instead of listing generic observations.
5. **Concise over filler:** If there is no pattern worth mentioning, do not fabricate one.
6. **Accurate over invented:** Every analytical result must be backed by concrete case content. **In the final report, include the {case_id} and the specific content from that case**case\{}_id and the specific content from that case} that supports the conclusion. Never fabricate. Each conclusion must have supporting case IDs and their specific content.
7. **Data-driven over impressions:** **You must actively collect and compute statistics**. Use the python\{}_repl\{}_tool to compute statistics on saved JSON data—means, percentages, distributions, etc. Avoid subjective terms like "significant" and "obvious"; use concrete numbers. Whenever a pattern or conclusion is found, quantify its prevalence.
8. **Full-coverage over sampling bias:** **When the total number of cases is fewer than 30, you must analyze *all* cases.** In the final report, state the actual number of cases and your coverage. If sampling, mark clearly at the start of each pattern "Based on Y sampled cases out of X total," to avoid misleading readers into thinking the sample reflects the whole.
9. **Objective description over subjective judgment:** **Do not output subjective evaluations or improvement suggestions.** Do not include sections like "capability boundaries and improvement suggestions," "summary of strengths/weaknesses," "areas to improve," etc. Only describe data- and case-based patterns. **Absolutely no capability-summary sections.**

**Report Structure Generated by the Reporter**

This section describes the expected report structure. Consider what information you need to complete it when designing your analysis strategy.

**Formatting Requirements**

- **Title:** A short declarative sentence stating which model analysis yields what conclusion, so

readers can grasp the point immediately.
- **TL; DR:**
- State the setting, conclusions, and key patterns in 3–5 sentences.
- **Data description** must clearly specify each case's benchmark and type (e.g., instruction-following, reasoning & STEM, agent, knowledge).
- **Analysis target** must be explicit (e.g., a specific model name).
- **Core conclusions** must be specific and supported by data and cases.
- For difficult-to-grasp summaries, include case examples to aid understanding.
- Use plain, concise language. Avoid words like "outstanding performance," "defect," or "fundamental weakness." It is acceptable to use "better," "good," or "worse."
- **Reference examples:**

```
case 1
User query: How does the model perform on cryptoMMLU?
Title: Crypto-MMLU evaluates fluid intelligence
TL; DR
Background:
- Fluid intelligence and crystallized intelligence are psychological concepts;
roughly, fluid depends on flexibility and speed, crystallized on knowledge.
- We observed: compared to SOTA (GPT-4O and Claude 3.5 Sonnet), Doubao's in-domain
capability is close, while OOD lags. Borrowing the terms above: crystallized is
close; fluid shows a noticeable gap.
Method & conclusions:
We build Crypto-MMLU by encrypting (encoding) the stem words of MMLU questions to
evaluate model ability. It is simple, tunable in difficulty, has a single variable
conducive to analysis, and connects current in-domain tasks with OOD tasks. On
Crypto-MMLU, we ran multiple experiments and found:
- On Crypto-MMLU, p6d7.rl29 vs. Claude 3.5 Sonnet: on the 0% encoding set, the
gap is about -3 pp; on the 100% encoding set, -44 pp. The difference is clear,
suggesting p6d7.rl29's OOD ability is notably lower than Claude 3.5 Sonnet.
- For p6d7.rl29, p6d7.sft29, p6d7.base, the drop from 0% to 100% encoding is ~-40
pp in all three stages---OOD capability is similar across stages.
- Adding in-distribution Crypto-MMLU data to SFT for a 3.3B model raises
100%-encoding accuracy by ~7 pp, still clearly below SOTA; this suggests limited
potential to lift ceilings via SFT alone for this pattern; core differences likely
originate from pretraining.

case 2
User query: What response characteristics appear in OpenAI's O-series models?
Title: Observed response traits of GPT-O1 and GPT-4O on omni3.6
TL; DR:
1) GPT-O1's output has three parts: Completion tokens, Reasoning tokens, and
Response tokens. Completion tokens are OpenAI's billable tokens; Reasoning tokens
are the hidden CoT tokens; Response tokens are the output content.
2) Observing GPT-O1's Completion tokens: more complex tasks consume more
tokens.  On omni3.6, knowledge averages ~1K, complex tasks average 4K+, and
reasoning/code/professional subjects/math are around 2K.
3) Comparing GPT-O1 and GPT-4O response tokens: GPT-O1's responses are clearly
longer. On \knowledge," O1 is 2.26× 4O; elsewhere ~1.3--1.6×.
4) Comparing Completion tokens: GPT-O1 consumes ~6--30× GPT-4O.

case 3
User query: How distilled are various models?
Title: Detecting distillation via prompt engineering
TL; DR:
1) Use cognitive jailbreak + prompting to gauge distillation from a reference
model [currently GPT].
2) Qwen and Dpsk show high levels of distillation, possibly more than Phi-4.
3) With essentially no explicit distillation, Doubao's self-awareness is below
Claude Stable; higher false positives.
4) Llama-3.1 may also have undergone some distillation.

case 4
```

```
User query: Analyze tool-use differences across models
Title: Tool-use behavior on SWEBench and Multi-SWEBench---Claude vs. Doubao
TL; DR:
In the CodeAgent framework, we analyze tool-use traces from SWEBench_Verified and
Multi-SWEBench to investigate reasons behind score differences, with a focus on
Claude-4 vs. Doubao-1.6. Observations include:
- Claude-4 uses near the 50-round limit, frequently testing its code via tools
(heavy use of execute_bash), and almost no hallucinated tool use.
- Doubao-1.6 uses far fewer rounds (often under 10), hallucinates tools (uses
nonexistent ones), and rarely tests its code via tools.
```

**Statistical Analysis Information**

- Total number of analyzed cases
- Number of summarized patterns and each pattern's share
- Separate descriptions by dimension
- Overall performance data (scores, success rates, etc.)
- Explicit comparisons with other models (must name the compared models)

**Detailed Pattern Descriptions**

Explain each pattern in detail using tables:

**Single-model analysis table format**

— case_id — prompt summary — answer — model prediction — score — analysis of cause
— pattern —
— — — — — — — — — — — — — — — —
— fff134bb-d7a5-47b1-baa9-4e372981275a — [prompt requirements] — [gold answer] — [model
response] — 0.2 — [detailed error analysis] — [pattern name] —

**Multi-model comparison table format**

— case_id — prompt summary — answer — model-1 prediction — model-2 prediction —
model-1 score — model-2 score — analysis of cause — pattern —
— — — — — — — — — — — — — — — —
— fff134bb-d7a5-47b1-baa9-4e372981275a — [prompt requirements] — [gold answer] — [model-1
response] — [model-2 response] — 0.2 — 0.8 — [comparative analysis] — [pattern name] —

**Output Requirements**

1. Deeply understand and **write the report with reference to the principles**; keep paragraphs compact; minimize extra breaks.
2. **State only facts**. No extensions, associations, or subjective evaluations. Do not output promotional summaries such as "how to optimize the model."
3. Always use the language specified by locale, with plain and objective style.
4. **Integrate the analyzer's raw findings**, keep detailed content intact, and avoid losing information due to re-summarization.
5. **Directly output the report**: The final output **must start with the report content**—begin with the title (e.g., \{}# Analysis Report for XX Model). Do **not** include any prefaces such as "Below is the analysis report" or "According to the results."
6. **Data-driven**: Each conclusion must be supported by at least **5 different** {**case_id**}scase\{}_ids}, with detailed explanations of how specific content ($\geq 50$ words each) supports it. Preserve statistics (shares, error counts, etc.).
7. **If there is not enough supporting data, do not invent patterns**; omit them directly.
8. **Explicitly quote** which specific part of the *model prediction* is incorrect.
9. **Avoid generalities**; provide concrete textual evidence.
10. **Each cited** {**case_id**} **must remain complete and accurate**case\{}_id must remain complete and accurate}; do not truncate or omit any characters.

# F  TOOLS

Table 6: Overview of tools for AGENT4WEAKNESS.

| Tool | Inputs | Purpose | Tag |
|------|--------|---------|-----|
| get evaluation info | *none* | Retrieve the set of available evaluation models and benchmarks. | Data acquisition |
| get ability tool | *none* | Return a Markdown table listing scores of multiple models across capability dimensions and benchmarks. | Data acquisition |
| get ability sota tool | *none* | Return a nested dictionary Dict[str, Dict[str, Any]]: the outer dict is keyed by capability-tree nodes (including overall score, specific capabilities, and associated benchmark names); each inner dict contains sota (model name) and score (numeric value). | Data acquisition |
| get benchmark descriptions tool | *none* | Return a string containing descriptive summaries for each benchmark. | Data acquisition |
| get significance tool | model | Using the specified model as the baseline, compute other models' score differences, percentage changes, improvements, and statistical significance relative to the baseline. | Data analysis |
| get ability by models tool | models | Return the scores of each model in the provided list. | Data acquisition |
| get ability by dimensions tool | dimensions | Return, following the hierarchical structure, all models' scores on the specified capability dimension(s). | Data acquisition |
| get models metrics tool | models, metrics | Return the requested metrics for the given models across all benchmarks. | Data acquisition |
| get benchmark metrics tool | benchmark, metrics | Return the requested metrics on the specified benchmark (for all available models). | Data acquisition |
| get benchmark description by dimension tool | dimensions | Return descriptions of all benchmarks under the given capability dimension(s). | Data acquisition |
| get rank by dimension tool | model, dimension | Return the rank of the given model on the specified capability dimension. | Data analysis |
| count token tools | string | Count tokens in the input string and return an integer. | Data acquisition |
| analyze model tiers tool | data, capability, alpha, delta score, delta d, enforce cd | Analyze performance differences and statistical significance among models on a capability; perform gap-aware tiering: models are grouped only when results are non-significant with small score gaps, small effect sizes, and (optionally) rank differences within a critical difference, thereby separating models with large gaps. | Data analysis |
| analyze ability correlations tool | data | Analyze correlations among capabilities, assessing whether pairs of abilities co-vary. | Data analysis |
| analyze correlation tool | list a, list b | Compute the Pearson correlation coefficient for two numeric lists and return a natural-language interpretation. | Data analysis |
| get capability tree | *none* | Return the capability tree as Markdown, including root and leaf nodes. | Data acquisition |
| get benchmark info | benchmark | Return case-field information and summary statistics for the specified benchmark. | Data acquisition |

*Continued on next page*

Table 6: Tool list and functions (continued)

| Tool | Inputs | Purpose | Tag |
|------|--------|---------|-----|
| get single case token estimate | model, benchmark, filter type, score threshold | Estimate the token count for a single case under the given settings. | Data acquisition |
| filter cases of single model | model, benchmark, num cases, filter type, score threshold | Filter and return cases for a single model according to the specified criteria. | Data acquisition |
| filter cases of models | models, benchmark, num cases, filter type, score threshold | Simultaneously filter cases for multiple models using the given criteria. | Data acquisition |
| get cases by pattern | model, benchmark, initial case count, filter type, score threshold | Automatically analyze all cases of the specified benchmark. | In-depth analysis |
| save important info | case ids, model, benchmark, save reason | Save the specified cases to disk for follow-up analysis. | Data acquisition |

# G SUPPLEMENTARY EXPERIMENTS AND DISCUSSIONS

## G.1 QUERY DESIGN

Queries Q1–Q3 represent three primary categories of user requirements for model evaluation: Q1 focuses on performance weaknesses, Q2 on resource consumption, and Q3 on behavioral deficiencies. For these general-purpose queries, AGENT4WEAKNESS is designed to generate comprehensive, multi-faceted reports. We elaborate on the specific aspects covered by each query type below:

- **Q1: Identifying Performance Weaknesses.** This requires AGENT4WEAKNESS to compare the performance of the target model against other models (including SOTA and series-specific counterparts) across multiple benchmarks. The analysis includes performance gaps, rankings, and tiering. It also assesses performance consistency across different benchmarks to identify potential "seesaw" effects (i.e., fluctuating capabilities).

- **Q2: Analyzing Resource Inefficiencies.** This requires AGENT4WEAKNESS to conduct a comprehensive analysis of the target model's performance, inference time, and token consumption relative to other models. This analysis considers the impact of benchmark difficulty on efficiency and examines the correlation between token usage and performance gains.

- **Q3: Detecting Behavioral Deficiencies.** This requires AGENT4WEAKNESS to identify undesirable behavioral patterns in the target model's responses that differ from those of other models and contribute to poor performance. Examples include evaluating its instruction-following capability or assessing the frequency and success rate of self-reflection within its outputs.

To validate the real-world relevance of these query types, we surveyed 51 LLM practitioners. Their needs for identifying model weaknesses aligned with these three categories, which accounted for 41.1%, 7.1%, and 51.3% (totaling 99.5%) of their reported queries, respectively. As illustrated in Figure X, this distribution confirms that our query categories effectively address dominant user concerns. To further demonstrate the flexibility and generalization of Agent4Weakness, we also introduce three additional, fine-grained queries: Q4, Q5, and Q6.

Table 7: Scores of the baselines and AGENT4WEAKNESS across 4 evaluation dimensions with a maximum score of 10. Avg denotes the average scores across the three queries on the same dimensions. The highest average score is highlighted in **bold**.

| Model | Method | Query | Requirement Fulfillment | Content Value | Factuality | Readability |
|---|---|---|---|---|---|---|
| GPT-5 | Direct QA | Q1 | 4.7 | 4.7 | 3.6 | 6.4 |
| | | Q2 | 6.8 | 5.0 | 7.2 | 8.0 |
| | | Q3 | 4.5 | 5.0 | 5.3 | 6.6 |
| | | Avg | 5.3 | 4.9 | 5.4 | 7.0 |
| | One-Agent | Q1 | 6.4 | 5.5 | 6.9 | 8.0 |
| | | Q2 | 6.5 | 5.3 | 7.8 | 8.2 |
| | | Q3 | 6.5 | 5.0 | 7.2 | 7.8 |
| | | Avg | 6.5 | 5.3 | 7.3 | 8.0 |
| | AGENT4WEAKNESS | Q1 | 8.7 | 9.1 | 9.3 | 8.4 |
| | | Q2 | 7.9 | 7.5 | 9.1 | 8.9 |
| | | Q3 | 7.5 | 7.1 | 8.5 | 8.1 |
| | | Avg | **8.0** | **7.9** | **9.0** | **8.5** |
| Gemini-2.5-pro | Direct QA | Q1 | 7.3 | 6.7 | 4.7 | 7.8 |
| | | Q2 | 7.6 | 6.5 | 7.3 | 7.6 |
| | | Q3 | 4.8 | 4.6 | 4.3 | 6.7 |
| | | Avg | 6.6 | 5.9 | 5.4 | 7.4 |
| | One-Agent | Q1 | 6.7 | 4.9 | 5.2 | 7.3 |
| | | Q2 | 7.8 | 4.7 | 7.3 | 8.3 |
| | | Q3 | 5.0 | 4.4 | 4.7 | 8.0 |
| | | Avg | 6.5 | 4.7 | 5.7 | 7.9 |
| | AGENT4WEAKNESS | Q1 | 8.7 | 7.2 | 8.0 | 8.3 |
| | | Q2 | 7.5 | 7.0 | 7.7 | 8.5 |
| | | Q3 | 8.5 | 7.2 | 7.5 | 8.6 |
| | | Avg | **8.2** | **7.1** | **7.7** | **8.5** |

Table 8: Scores of the baselines and AGENT4WEAKNESS across 4 evaluation dimensions with a maximum score of 10. Avg denotes the average scores across the three queries on the same dimensions. The highest average score is highlighted in **bold**.

| Method | Time | Input Tokens | Output Tokens | RF | CV | F | R |
|---|---|---|---|---|---|---|---|
| Direct QA | 21.7s | $109,107.9$ | $1,912.3$ | 6.9 | 6.6 | 7.3 | 8.3 |
| One-Agent | 170.2s | $105,105,026.0$ | $6,347.2$ | 7.0 | 4.5 | 6.4 | 5.0 |
| AGENT4WEAKNESS | 246.3s | $107,013,483.3$ | $15,336.7$ | 8.9 | 8.7 | 8.1 | 8.7 |
| Human Annotator | 30h | - | - | 10.0 | 9.0 | 10.0 | 9.5 |

## G.2 RUNNING AGENT4WEAKNESS WITH OTHER MODELS

In addition to Claude-Opus-4.1-thinking, which was used in our main experiments, we also evaluate GPT-5 (OpenAI, 2025) and Gemini-2.5-pro (Google, 2025b). A comparison of their performance is presented Table 7. For consistency, we continue to use Claude-Opus-4.1-thinking Anthropic (2025) as the evaluator. We observe that AGENT4WEAKNESS still consistently and significantly outperforms the baseline across all four dimensions.

## G.3 EFFICIENCY OF AGENT4WEAKNESS

We begin by comparing the time and token consumption of AGENT4WEAKNESS, the baseline method, and professional human evaluators on Q3. The Q3 task is selected because it demands extensive observation of model responses across multiple evaluation sets, making it the most resource-intensive of all queries. For the human-annotated results, we previously commissioned professional LLM evaluators to generate reports for the Q3 task across eight models; the annotation time is an approximate statistic, and these reports are subsequently scored by Claude-Opus-4.1-thinking.

While the baseline method consumes fewer resources, its unacceptably low scores across the four dimensions of Requirement Fulfillment, Content Value, Factuality, and Readability render it inade-

quate. Conversely, manual annotation by experts is excessively time-consuming and lacks scalability. Furthermore, human analysis is inherently constrained by individual perspectives, often causing evaluators to overlook or miss specific model deficiencies during large-scale instance analysis. This limitation frequently results in the Content Value dimension scoring lower than the others. Therefore, we argue that the resource consumption of AGENT4WEAKNESS is necessary and justified. It represents a step toward the automated, flexible, and high-quality discovery of model weaknesses.

## G.4 ROBUSTNESS OF AGENT4WEAKNESS

To assess the robustness of AGENT4WEAKNESS, we conduct 5 independent runs for each query, analyzing the deficiencies of GPT-5-high. To mitigate potential biases from LLM-based scoring, we employ human evaluation.

The results are presented in Table 9. We further quantify the stability of AGENT4WEAKNESS by calculating the **average (Avg)** and **standard deviation (Std)** across the 5 runs for each metric. The results demonstrate exceptional consistency: the standard deviations are extremely low across all queries and dimensions. Notably, the highest observed standard deviation is merely $0.89$ (for Q3-Factuality), with most metrics exhibiting an SD of $\approx 0.5$ or less (e.g., Q1-Readability and Q2-Content Value show an SD of $0.00$). This low variance quantitatively confirms that AGENT4WEAKNESS produces stable and reliable results, minimizing random fluctuations across independent runs.

Table 9: Comparison of AGENT4WEAKNESS with five runs across 4 evaluation dimensions (RF: Requirement Fulfillment, CV: Content Value, F: Factuality, R: Readability), with a maximum score of $10$. We report scores for each run, along with the **Average (Avg)** and **Standard Deviation (Std)** across runs to demonstrate robustness.

| | Q1 | | | | Q2 | | | | Q3 | | | |
|------|------|------|------|------|------|------|------|------|------|------|------|------|
| | RF | CV | F | R | RF | CV | F | R | RF | CV | F | R |
| run1 | 8.0 | 8.0 | 9.0 | 8.0 | 9.0 | 9.0 | 8.0 | 9.0 | 8.0 | 8.0 | 8.0 | 9.0 |
| run2 | 9.0 | 8.0 | 8.0 | 8.0 | 10.0 | 9.0 | 8.0 | 8.0 | 9.0 | 9.0 | 8.0 | 9.0 |
| run3 | 9.0 | 8.0 | 8.0 | 8.0 | 10.0 | 9.0 | 8.0 | 8.0 | 9.0 | 8.0 | 10.0 | 9.0 |
| run4 | 9.0 | 9.0 | 8.0 | 8.0 | 10.0 | 9.0 | 8.0 | 9.0 | 9.0 | 9.0 | 8.0 | 10.0 |
| run5 | 9.0 | 8.0 | 8.0 | 8.0 | 10.0 | 9.0 | 9.0 | 8.0 | 9.0 | 9.0 | 8.0 | 9.0 |
| **Avg** | 8.6 | 8.2 | 8.2 | 8.0 | 9.8 | 9.0 | 8.2 | 8.4 | 8.6 | 8.6 | 8.4 | 9.2 |
| **Std** | 0.50 | 0.45 | 0.45 | 0.00 | 0.45 | 0.00 | 0.45 | 0.55 | 0.50 | 0.55 | 0.89 | 0.45 |

## G.5 RANGE OF MODEL WEAKNESSES

We classify model weaknesses into two distinct categories: objective and subjective (Song et al., 2025). (*i*) Objective weaknesses encompass capability deficits (e.g., inferior performance or lower rankings on specific datasets compared to other models) and behavioral flaws (e.g., severe hallucinations leading to incorrect responses). To enable AGENT4WEAKNESS to identify these objective issues, we provide evaluation data from other models and ground-truth answers as references. Furthermore, our main and analytical experiments quantitatively demonstrate that AGENT4WEAKNESS accurately identifies these limitations, significantly outperforming baselines. (*ii*) Subjective weaknesses refer to aspects such as a model's perceived unsuitability for a particular task or preferences regarding its linguistic style. Due to the absence of standardized evaluation metrics (Song et al., 2025), subjective weaknesses fall outside the scope of this paper.

