# OpenReview forum: "Beyond Score: A Multi-Agent System to Discover Capability and Behavioral Weaknesses in LLMs"
_ICLR.cc/2026/Conference — ICLR 2026 Conference Withdrawn Submission_

### Official Review · Reviewer_4zmv · 2025-10-30

**Soundness:** 3
**Presentation:** 3
**Contribution:** 2
**Rating:** 6
**Confidence:** 4

**Summary:**

The paper introduces AGENT4WEAKNESS, a multi-agent system designed to address the limitations of existing methods for discovering capability and behavioral weaknesses in large language models (LLMs). Current weakness discovery approaches are characterized by insufficient comparison, often failing to analyze statistical significance and confidence of performance differences, and inflexible evaluation, being restricted to fixed perspectives rather than adapting to user-specific requirements. AGENT4WEAKNESS aims to provide richer statistical insights and generate customized evaluation reports through collaborative agents and specialized tools.
The core methodology of AGENT4WEAKNESS is structured around a multi-agent workflow consisting of a Planner, an Analyzer, and a Reporter, leveraging comprehensive evaluation data. Experimental results demonstrate that reports generated by AGENT4WEAKNESS show a significant improvement of 2.6 out of 10 points across four evaluation dimensions (Requirement Fulfillment, Content Value, Factuality, and Readability) compared to a direct answering baseline, with high consistency with human evaluations (Pearson r=0.801, Spearman \rhoρ\rhoρ=0.944). Notably, it achieves a 3.4-point improvement in Content Value, signifying richer analyses, and a 3.4-point improvement in Requirement Fulfillment, indicating its flexibility. Furthermore, models guided by weaknesses discovered through AGENT4WEAKNESS show an average performance improvement of 3.7 points, underscoring its practical utility.

**Strengths:**

1. The paper squarely tackles the challenge of LLM weakness discovery, a critical need as models grow more complex. The authors clearly motivate that current eval methods lack substantive comparisons (no statistical rigor) and flexibility to meet different analysis needs.
2. The proposed Agent4Weakness framework is innovative in using a multi-agent collaboration for evaluation. It combines a Planner, Analyzer, and Reporter agent with specialized roles. This design allows the system to break down the user’s query, fetch and compute detailed statistics, and compile findings into a coherent report. The use of external tools (for data analysis and statistical testing) within the agent workflow is a strong point, as it grounds the LLM’s analysis in solid quantitative evidence rather than just its internal knowledge. This represents a creative extension of chain-of-thought prompting into a tool-augmented, multi-step evaluation process.
3. Agent4Weakness produces much more informative and accurate analyses than the baseline. The strong performance across multiple dimensions underscores the efficacy of the approach. Furthermore, the authors show that LLM-based scoring of the reports correlates strongly with human evaluation (Pearson r ≈ 0.80, Spearman ρ ≈ 0.94),  which suggests the evaluation criteria were meaningful and actually reflect human-perceived quality.
4. The system doesn’t just produce higher-level summaries – it can pinpoint specific weaknesses with verifiable accuracy. Such detailed weakness analysis is a clear strength over traditional evaluations that would simply report an average score.

**Weaknesses:**

1. High Complexity and Resource Requirements: The proposed system is quite complex, involving three coordinated agent roles and multiple tool integrations. Running Agent4Weakness requires a powerful backbone LLM and a large set of evaluation data in memory. This complexity could pose practical challenges. For example, orchestrating multi-agent prompts and tool calls may be slow or expensive compared to a single-pass evaluation.
2. The paper primarily compares Agent4Weakness to a direct answering baseline. While this highlights the advantage of the structured approach, the baseline is relatively rudimentary. It does not, for instance, use any chain-of-thought or tools at all. In other words, an intermediate baseline (say, a single-agent chain-of-thought using the same tools, but without specialized roles) could help attribute the improvements more precisely. As it stands, the evaluation convincingly shows superiority to an “uninformed” baseline, but not necessarily to any sophisticated alternative.
3. The paper demonstrates that Agent4Weakness’s outputs correlate well with human judgments of quality, but it doesn’t compare against human-written analysis of model weaknesses. Of course, expecting the authors to produce human-written reports for all cases is impractical, so this is more of an observation than a strict criticism.

**Questions:**

1. You chose a direct-answer baseline for comparison, arguing that prior specialized pipelines aren’t as flexible. Nonetheless, have you considered comparing Agent4Weakness to a simpler single-agent chain-of-thought approach using tools? For example, one could prompt a single LLM with something like: “Here is all the data, think step by step to analyze weaknesses…” and allow it to call the same tools (sort of an ablation of the multi-agent structure). This might isolate the benefit of having distinct Planner/Analyzer/Reporter roles.
2. The multi-agent pipeline with external tools sounds computationally heavy (multiple prompt exchanges, tool calls, etc.). Do you have a sense of the runtime or cost overhead of Agent4Weakness compared to a direct evaluation? For instance, how long does it take to generate a full report for one model’s weaknesses, and could this be a bottleneck if evaluating many models continuously?
3. You showed a compelling result that providing the model with the identified weaknesses and suggestions can improve its performance via prompt adjustments. Do you envision a more automated integration of Agent4Weakness into the model development loop? For example, could the reports be used to inform fine-tuning data generation or to create adversarial test cases for continuous improvement?

---

> ### Author Response · Authors · 2025-11-17
> **Response to Reviewer 4zmv**
>
> Thank you for recognizing our work. Below are our responses to your concerns. We have marked the revisions in **blue** in the manuscript.
>
> **W1.** High Complexity and Resource Requirements. ***​(​Appendix G.3 in Revision)***
>
> * We argue that the resource consumption of Agent4Weakness is a necessary trade-off, compared with the simple baselines and human annotators.
> * We first present a comparison of the time and token consumption for Agent4Weakness, the baselines, and professional human annotators, specifically on the Q3 task.
>   * We select Q3 as it is the most resource-intensive query, requiring extensive observation of model responses across multiple evaluation datasets.
>   * The "One-Agent" baseline utilizes the identical set of tools available to Agent4Weakness but operates as a single agent without specialized role differentiation.
>   * For the "Human Annotators" results, we commissioned professional LLM evaluators to generate reports for the Q3 task across 8 models. The 30-hour duration is an approximation of the time required.
>
> | **Method** | **Time** | **Input Tokens** | **Output Tokens** | **Requirement Fulfillment** | **Content Value** | **Factuality** | **Readability** |
> | :--- | :---: | :---: | :---: | :---: | :---: | :---: | :---: |
> | Direct QA | 21.7s | 109,107.9 | 1,912.3 | 6.9 | 6.6 | 7.3 | 8.3 |
> | One-Agent | 170.2s | 105,105,026.0 | 6,347.2 | 7.0 | 4.5 | 6.4 | 5.0 |
> |Agent4Weakness | 246.3s | 107,013,483.3 | 15,336.7 | 8.9 | 8.7 | 8.1 | 8.7 |
> | Human Annotator | 30h | - | - | 10.0 | 9.0 | 10.0 | 9.5 |
>
>
> * While the **baselines** consume fewer resources, their **low scores** across the four quality dimensions are unacceptable.
> * Conversely, ​**human evaluation, while high-quality, is prohibitively time-consuming and not scalable**​. Furthermore, manual analysis is inevitably limited by individual perspectives, potentially causing evaluators to overlook certain model weaknesses across large-scale instances. This limitation may explain why the human-annotated "Content Value" is slightly lower than its other scores.
> * Therefore, we argue that the resource consumption of Agent4Weakness is a necessary trade-off. We posit that it represents a significant step toward **automated, flexible, and high-quality** identification of model weaknesses.
>
> **W2.** Simple Baselines. ***​(Section 3 in Revision)***
>
> * Agent4Weakness consistently and significantly outperforms the One-Agent baseline, demonstrating the clear benefit of employing distinct Planner, Analyzer, and Reporter roles.
> * Specifically, we conduct a comparison between Agent4Weakness and a **One-Agent** baseline. This baseline utilizes the identical set of tools available to Agent4Weakness but operates as a single agent without specialized role differentiation.
>
> | Method | Query | Requirement Fulfillment | Content Value | Factuality | Readability |
> | :--- | :--- | :--- | :--- | :--- | :--- |
> | Direct QA | Q1 | 6.7 | 5.7 | 6.4 | 7.9 |
> | Direct QA | Q2 | 3.7 | 3.0 | 3.3 | 6.4 |
> | Direct QA | Q3 | 6.9 | 6.6 | 7.3 | 8.3 |
> | Direct QA | Avg | 5.8 | 5.1 | 5.7 | 7.5 |
> | One-Agent | Q1 | 6.7 | 5.7 | 6.0 | 7.3 |
> | One-Agent | Q2 | 6.7 | 5.5 | 5.0 | 6.4 |
> | One-Agent | Q3 | 7.0 | 4.5 | 6.4 | 5.0 |
> | One-Agent | Avg | 6.5 | 5.2 | 5.8 | 6.2 |
> |Agent4Weakness | Q1 | 8.8 | 8.3 | 9.7 | 7.5 |
> |Agent4Weakness | Q2 | 9.9 | 8.6 | 8.0 | 8.4 |
> |Agent4Weakness | Q3 | 8.9 | 8.7 | 8.1 | 8.7 |
> |Agent4Weakness | Avg | **8.8** | **8.4** | **8.6** | **8.1** |
>
> **W3.** Comparison with Human-Written Reports. ***​(​Appendix G.3 in Revision)***
>
> Please see our response to W1.
>
> **Q1: ​**Compare Agent4Weakness to a simpler single-agent chain-of-thought approach using tools. ***​(Section 3 in Revision)***
>
> Please see our response to W2.
>
> **Q2: ​**Computational overhead of Agent4Weakness. ***​(​Appendix G.3 in Revision)***
>
> * Regarding computational overhead, please refer to our detailed response to ​**W1**​.
> * Additionally, if the task requires evaluating the weaknesses of multiple models, the report generation process for each model can be executed in parallel, thereby mitigating scalability concerns.
>
> **Q3.** Integrating Agent4Weakness into Model Development.
>
> * We appreciate this valuable suggestion and plan to pursue it as ​**future work**​.
>   * Specifically, we intend to leverage the weaknesses identified by Agent4Weakness for ​**targeted data synthesis**​.
>   * Furthermore, we will explore using both the discovered negative (weakness) and positive (strength) cases as preference data for **reinforcement learning**​**​ ​**to further enhance model performance.

---

### Official Review · Reviewer_DEcy · 2025-11-01

**Soundness:** 4
**Presentation:** 2
**Contribution:** 2
**Rating:** 4
**Confidence:** 4

**Summary:**

This paper introduces AGENT4WEAKNESS, a multi-agent system designed to overcome the limitations of current LLM evaluation methods, specifically their "insufficient comparison" and "inflexible evaluation". The framework leverages multi-agent collaboration (a Planner, Analyzer, and Reporter) and a suite of specialized statistical and analytical tools to generate in-depth evaluation reports based on flexible user queries. Rather than just reporting scores, the system performs both capability analysis (assessing statistical significance of performance gaps) and behavioral analysis (mining reasoning patterns from Chain-of-Thought data). The authors demonstrate through extensive experiments that reports from AGENT4WEAKNESS are significantly higher in quality than a strong baseline.

**Strengths:**

-  The paper's primary innovation is reframing LLM evaluation from a static, fixed-pipeline task to a flexible, systematic process. An agentic workflow with multi agents are used.
- The technical quality of the work is high. The system is thoughtfully designed, decomposing the complex task of "weakness discovery" into distinct, manageable agent roles and tool families. In general the design of the system makes sense.

**Weaknesses:**

```Lack of evidence of the benefit from the report ```

I am concerned whether the report is helpful given that each model is unique. Without knowing the internals or the properties of the model, it would be hard to assess whether the report is helpful. One example given in the paper is that AGENT4WEAKNESS identifies that the reasoning process of DeepSeek-V3.1 on AIME2025 questions is disorganized and suggests using markers such as “### Step 1” to structure the reasoning and adding verification of intermediate results after each step. This particular example can be problematic because for some models, adding additional structure or verification may force model to reason out of distribution. This may lead to decreased performances.

```Nature of Model "Improvement" is in context only```

The 3.7-point performance improvement (Section 4.3) is a strong result, but it appears to be achieved by feeding the system's analysis and suggestions back into the model as part of the prompt. This demonstrates an improvement in in-context learning or prompt adherence based on the analysis, which is different from a permanent improvement to the base model (e.g., via fine-tuning). While still valuable, this distinction should be made clearer, as the current phrasing could imply a more fundamental model enhancement.

**Questions:**

- Did you experiment with other SOTA models (like GPT-5 or Gemini-2.5-pro) to run the agents themselves? How sensitive is the final report quality ("Content Value" and "Factuality") to the choice of the underlying model powering the Planner, Analyzer, and Reporter?

---

> ### Author Response · Authors · 2025-11-17
> **Response to Review DEcy**
>
> Thank you for recognizing our work. Below are our responses to your concerns. We have marked the revisions in **blue** in the manuscript.
>
> **W1.** Lack of evidence of the benefit from the report. ***(Section 4.3 in Revision)***
>
> * We ensure the benefit of the report by inputting comprehensive evaluation data and designing the prompts of Agent4Weakness.
> * To ensure the report's utility, we ​**collect comprehensive evaluation information as the input of Agent4Weakness**​, such as the performance of the models on diverse benchmarks and comparative results from other models. This methodology ensures that the identified weaknesses are objective and not anomalous.
> * Moreover, we emphasize in the prompts of Agent4Weakness that ​**all conclusions and recommendations must be summarized in empirical data**​, avoiding speculation.
>   * Regarding your example: Agent4Weakness noted that DeepSeek-V3.1's reasoning process on AIME2025 tasks was disorganized. It recommended using structured markers (e.g., "### Step 1") and adding intermediate verification after each step.
>   * This recommendation was not speculative. Agent4Weakness derived this insight by acquiring and comparing DeepSeek-V3.1's successful cases against its failures. It observed that ​**in the few instances where DeepSeek-V3.1 adopted this structured pattern, it achieved high scores**​.
>   * The resulting average performance increase of 3.7 points further validates the efficacy of this data-grounded suggestion.
> * This highlights the core strength of Agent4Weakness: its ability to ​**perform fine-grained analysis of large-scale model responses**​, identify behavioral deficits, and provide actionable, evidence-based recommendations.
>
> **W2.** Nature of Model "Improvement" is in context only. ***(Lines 32 and 98-99 in Revision)***
>
> * We have revised the manuscript to explicitly clarify that the observed improvement is achieved **in-context** rather than model training.
>
> **Q1.** Sensitivity of final report quality to the underlying model. ***(Appendix G.2 in Revision)***
>
> * Agent4Weakness **retains high performance across models** and still consistently and significantly outperforms the baseline across all four dimensions.
> * Specifically, we compare the performance of Claude-Opus-4.1-thinking with GPT-5 and Gemini-2.5-pro. The results are presented below.
>
> | **Model** | **Method** | **Query** | **Requirement Fulfillment** | **Content Value** | **Factuality** | **Readability** |
> | :--- | :--- | :--- | :--- | :--- | :--- | :--- |
> | Claude-Opus-4.1-thinking | Direct QA | Q1 | 6.7 | 5.7 | 6.4 | 7.9 |
> | Claude-Opus-4.1-thinking | Direct QA | Q2 | 3.7 | 3.0 | 3.3 | 6.4 |
> | Claude-Opus-4.1-thinking | Direct QA | Q3 | 6.9 | 6.6 | 7.3 | 8.3 |
> | Claude-Opus-4.1-thinking | Direct QA | Avg | 5.8 | 5.1 | 5.7 | 7.5 |
> | Claude-Opus-4.1-thinking | Agent4Weakness | Q1 | 8.8 | 8.3 | 9.7 | 7.5 |
> | Claude-Opus-4.1-thinking | Agent4Weakness | Q2 | 9.9 | 8.6 | 8.0 | 8.4 |
> | Claude-Opus-4.1-thinking | Agent4Weakness | Q3 | 8.9 | 8.7 | 8.1 | 8.7 |
> | Claude-Opus-4.1-thinking | Agent4Weakness | Avg | **9.2** | **8.5** | **8.6** | **8.2** |
> | GPT-5 | Direct QA | Q1 | 4.7 | 4.7 | 3.6 | 6.4 |
> | GPT-5 | Direct QA | Q2 | 6.8 | 5.0 | 7.2 | 8.0 |
> | GPT-5 | Direct QA | Q3 | 4.5 | 5.0 | 5.3 | 6.6 |
> | GPT-5 | Direct QA | Avg | 5.3 | 4.9 | 5.4 | 7.0 |
> | GPT-5 | Agent4Weakness | Q1 | 8.7 | 9.1 | 9.3 | 8.4 |
> | GPT-5 | Agent4Weakness | Q2 | 7.9 | 7.5 | 9.1 | 8.9 |
> | GPT-5 | Agent4Weakness | Q3 | 7.5 | 7.1 | 8.5 | 8.1 |
> | GPT-5 | Agent4Weakness | Avg | **8.0** | **7.9** | **9.0** | **8.5** |
> | Gemini-2.5-pro | Direct QA | Q1 | 7.3 | 6.7 | 4.7 | 7.8 |
> | Gemini-2.5-pro | Direct QA | Q2 | 7.6 | 6.5 | 7.3 | 7.6 |
> | Gemini-2.5-pro | Direct QA | Q3 | 4.8 | 4.6 | 4.3 | 6.7 |
> | Gemini-2.5-pro | Direct QA | Avg | 6.6 | 5.9 | 5.4 | 7.4 |
> | Gemini-2.5-pro | Agent4Weakness | Q1 | 8.7 | 7.2 | 8.0 | 8.3 |
> | Gemini-2.5-pro | Agent4Weakness | Q2 | 7.5 | 7.0 | 7.7 | 8.5 |
> | Gemini-2.5-pro | Agent4Weakness | Q3 | 8.5 | 7.2 | 7.5 | 8.6 |
> | Gemini-2.5-pro | Agent4Weakness | Avg | **8.2** | **7.1** | **7.7** | **8.5** |

---

### Official Review · Reviewer_S6xB · 2025-11-01

**Soundness:** 2
**Presentation:** 2
**Contribution:** 2
**Rating:** 4
**Confidence:** 4

**Summary:**

This paper proposes AGENT4WEAKNESS, a multi-agent system designed to uncover capability and behavioral weaknesses of large language models beyond numeric evaluation scores. The framework coordinates three agents, Planner, Analyzer, and Reporter, equipped with 22 analytical tools to generate structured weakness reports from existing benchmark results. Tested on 8 models and 27 benchmarks, the system improves report quality and aligns closely with human judgments, demonstrating its effectiveness in revealing interpretable model weaknesses.

**Strengths:**

1. Well-motivated framework for deeper LLM evaluation.
The paper clearly identifies the limitations of traditional benchmark-based evaluation, which focuses only on numeric scores, and introduces a multi-agent framework (AGENT4WEAKNESS) that systematically analyzes capability and behavioral weaknesses of LLMs beyond accuracy metrics. This motivation is timely and addresses a meaningful gap in current evaluation practices.

2. Comprehensive and well-structured system design.
The proposed Planner–Analyzer–Reporter architecture is conceptually clear and technically coherent. Each agent has distinct responsibilities, and the framework integrates 22 analytical tools for statistical testing, capability gap detection, and behavioral pattern mining. The design effectively demonstrates how multi-agent coordination can enable flexible, user-driven evaluation.

3. Empirical validation.
The system is tested on 8 representative LLMs and 27 public benchmarks, covering reasoning, factuality, and safety. Alignment of human evaluation and model-based score confirm the validity of the automatically generated reports.

**Weaknesses:**

1. Limited technical innovation.
The framework mainly assembles existing components in the agentic working pipeline, including multi-agent role decomposition, tool invocation without introducing novel algorithms or coordination mechanisms.

2. Insufficient diversity of user queries.
The evaluation covers only three representative user requests (Q1–Q3), which cannot fully demonstrate the framework’s flexibility or generalization across different analysis needs. Broader testing on varied and real-world queries would be needed to justify the claimed adaptability and scalability of AGENT4WEAKNESS.

3. Limited ablation and component analysis.
The paper provides little insight into the contribution of each component (22 analytical tools and three agent roles). The ablation results shown in Table 4 remain coarse-grained and do not reveal which tools or interactions most affect report quality. A more detailed analysis would strengthen the causal understanding of the framework’s performance.

4. Lack of formal formulation.
While the overall system architecture (Figure 2) conveys the high-level workflow, the paper lacks a clear and formalized description of how the multi-agent coordination operates in practice. Key elements such as the data flow between the Planner, Analyzer, and Reporter agents, the intermediate representations of benchmark results, and the decision rules for tool invocation are only described narratively.

**Questions:**

1. In Section 3.1, the author first  mentions “a detailed list of 104 models” but later states that “the evaluation results of 8 representative models”, Could the authors clarify whether and how the 104 models are used in your experiments?

2. Only three user requests (Q1–Q3) are defined to test the framework. How do you make sure these requests are representative and cover the user needs?

---

> ### Author Response · Authors · 2025-11-17
> **Response to Reviewer S6xB**
>
> Thank you for recognizing our work. Below are our responses to your concerns. We have marked the revisions in **blue** in the manuscript.
>
> **W1.​** Limited technical innovation.​ ***(Lines 75-85, 89-100)***
>
> * We would like to highlight the following novel contributions:
>   * **Conceptual Novelty:** The primary innovation of our work is the development of a ​**more effective, in-depth, and scalable methodology for diagnosing model weaknesses**​. Unlike conventional benchmarks, which provide simple quantitative scores, Agent4Weakness seeks a deeper, qualitative understanding of a model's *true* deficiencies. We argue this diagnostic capability is crucial for the advancement of the field.
>   * **Technical and Practical Novelty:** At the implementation level, we systematically explore the specific design choices essential for our agent's effectiveness, including the integration of ​**roles, tools, and memory**​. Furthermore, we demonstrate the significant practical utility of our method: the diagnostic reports generated by Agent4Weakness can be directly leveraged as feedback (e.g., in prompts) to ​**measurably improve the model's performance**​.
>   * **Broader Impact:** Finally, our work advocates for a crucial shift in evaluation paradigms. We aim to move the community beyond a sole reliance on benchmark leaderboards, emphasizing the need for methods that facilitate a ​**more fundamental understanding of model behaviors and limitations**​.
>
> **W2.​** Insufficient diversity of user queries. ***​(Section 3 in Revision)***
>
> * Our agent is designed to **handle arbitrary queries**​**about model weakness** and maintains robust performance across them. The use of fixed queries in the main paper was primarily due to page limitations. To address this concern, we include  experiments **on three additional queries** to demonstrate our method's effectiveness on a more diverse set of queries.
> * Explanation of Q1-Q3 selection: Q1-Q3 represent ​**three primary categories of user requirements**. For such general queries, Agent4Weakness is designed to provide comprehensive, multi-faceted reports (i.e., high "Content Value"). We elaborate on the diverse aspects covered by these queries:
>   * **Q1 (Performance Weakness):** Requires comparing the target model against SOTA and series-specific models on multiple benchmarks, analyzing performance gaps, rankings, and potential performance trade-offs (i.e., "seesaw phenomenon").
>   * **Q2 (Efficiency Weakness):** Involves a comprehensive analysis of performance in conjunction with time and token consumption, considering the impact of benchmark difficulty on efficiency and the relationship between token usage and performance gains.
>   * **Q3 (Behavioral Weakness):** Involves identifying specific performance-degrading patterns that differ from other models, such as poor instruction following, or the frequency and success rate of self-reflection.
> * To validate the representativeness of these queries, we survey **51 ​LLM practitioners**​. These three categories (Q1, Q2, Q3) account for **41.1%,  7.1%​, and 51.3%** of their needs, respectively, covering **99.5%** of real-world use cases.
> * To further demonstrate the flexibility and generalization of Agent4Weakness, we supplement our experiments with three new, more specific queries (Q4, Q5, Q6) as fine-grained variations of Q1-Q3.
>   * **Q4:** "Analyze whether this model exhibits instruction-following failures, and under what conditions these failures are most severe."
>   * **Q5:** "Analyze the deficiencies exhibited by the model in its reflection behavior."
>   * **Q6:** "Analyze the relationship between the capabilities of the model and its Best-of-N capability upper bound."
>
> | Method | Query | Requirement Fulfillment | Content Value | Factuality | Readability |
> | :--- | :--- | :--- | :--- | :--- | :--- |
> | Direct QA | Q1 | 6.7 | 5.7 | 6.4 | 7.9 |
> | Direct QA | Q2 | 3.7 | 3.0 | 3.3 | 6.4 |
> | Direct QA | Q3 | 6.9 | 6.6 | 7.3 | 8.3 |
> | Direct QA | Q4 | 5.0 | 3.0 | 4.3 | 7.3 |
> | Direct QA | Q5 | 4.3 | 2.8 | 3.0 | 5.3 |
> | Direct QA | Q6 | 7.5 | 6.0 | 4.5 | 6.5 |
> | Direct QA | Avg | 5.7 | 4.5 | 4.8 | 6.7 |
> |Agent4Weakness | Q1 | 8.8 | 8.3 | 9.7 | 7.5 |
> |Agent4Weakness | Q2 | 9.9 | 8.6 | 8.0 | 8.4 |
> |Agent4Weakness | Q3 | 8.9 | 8.7 | 8.1 | 8.7 |
> |Agent4Weakness | Q4 | 8.7 | 7.9 | 8.4 | 7.7 |
> |Agent4Weakness | Q5 | 8.2 | 8.1 | 8.2 | 8.2 |
> |Agent4Weakness | Q6 | 8.5 | 8.5 | 9.2 | 8.3 |
> |Agent4Weakness | Avg | **8.8** | **8.4** | **8.6** | **8.1** |

---

> ### Author Response · Authors · 2025-11-17
>
> **W3. ​**Limited ablation and component analysis.​ ***(Section 4.2 in Revision)***
>
> * We conduct further experiments and present the following conclusions:
>   * **Agent Interactions:**
>     * Ablating the planner most significantly impacted Requirement Fulfillment and Content Value.
>     * Ablating the reporter resulted in comprehensive performance degradation across all four evaluation dimensions.
>   * **Roles:** Ablating the agent roles led to a performance decrease across all dimensions, confirming their crucial contributions, especially to Content Value and Factuality.
>   * **Tools:**
>     * Ablating the analysis tools primarily impacted Factuality.
>     * Ablating the in-depth tools showed the largest adverse effect on Content Value and Factuality, particularly for handling difficult problems, such as Q3.
> * Experimental Setup and Results: In addition to the original ablations on roles and tools, we have now included ablations for the **Planner** and **Reporter** agents. The **Analyzer** agent is essential for data retrieval, making its complete removal infeasible. Therefore, our original ablation of "analysis tools" (specifically the *data analysis*​*​ tool* and ​*deep analysis tool*​) effectively serves as an ablation of the Analyzer's advanced capabilities. We also add a specific ablation for **the ​**​​***deep analysis tool***​. The results are shown below (RF: Requirement Fulfillment, CV: Content Value, F: Factuality, R: Readability).
>
> | **Method** | **Q1** | **Q1** | **Q1** | **Q1** | **Q2** | **Q2** | **Q2** | **Q2** | **Q3** | **Q3** | **Q3** | **Q3** |
> | :--- | :---: | :---: | :---: | :---: | :---: | :---: | :---: | :---: | :---: | :---: | :---: | :---: |
> | **Method** | **RF** | **CV** | **F** | **R** | **RF** | **CV** | **F** | **R** | **RF** | **CV** | **F** | **R** |
> | Agent4Weakness | **8.8** | **8.3** | **9.7** | **7.5** | **9.9** | **8.6** | **8.0** | **8.4** | **8.9** | **8.7** | **8.1** | **8.7** |
> | Ablating roles | 8.3 | 7.4 | 8.6 | 6.7 | 7.9 | 8.0 | 7.7 | 6.1 | 4.6 | 1.6 | 1.6 | 6.1 |
> | Ablating analysis tools | 7.9 | 7.0 | 8.3 | 7.4 | 9.3 | 8.4 | 7.9 | 7.7 | 5.3 | 4.3 | 2.0 | 6.0 |
> | Ablating in-depth tools | 8.8 | 8.3 | 9.7 | 7.5 | 9.9 | 8.6 | 8.0 | 8.4 | 6.0 | 4.5 | 4.2 | 7.5 |
> | Ablating Planner | 6.5 | 6.1 | 6.5 | 7.4 | 8.0 | 6.4 | 7.3 | 6.0 | 5.6 | 2.0 | 6.0 | 6.2 |
> | Ablating Reporter | 7.2 | 6.5 | 6.8 | 6.3 | 6.7 | 5.0 | 6.6 | 5.8 | 4.8 | 3.0 | 5.0 | 5.7 |
>
> **W4. ​**Lack of formal formulation.​ ***(Section 2 in Revision)***
>
> * We have added the following formalizations as requested:
>   * The data flow between agents is already presented in ​**Equations 2.4-2.6**​, so we add the explicit reference in lines 174-177.
>   * For the intermediate representations of benchmark results, which was previously described in lines 152-153 for readability, we  now add a formal definition in ​**Equations 2.2-2.3**​.
>
>   $\mathcal{D}_{\text{eval}} = (\mathcal{D}_{\text{raw}}, \mathcal{D}_{\text{stat}})$
>
>   $\mathcal{D}_{\text{raw}} = \bigcup_{b \in \mathcal{B}} \left\{ (q_i, a_i^*, \{r_{m,i}\}_{m \in \mathcal{M}}) \mid i \in \mathcal{I}_b \right\}$
>
>   $\mathcal{D}_{\text{stat}} = \{ \text{stat}_k(m, b) \mid m \in \mathcal{M}, b \in \mathcal{B}, k \in \{1, \dots, K\} \}$
>
>   * We also formalize the decision rules for tool invocation in ​**lines 186-189 and lines 213-214**​.
>
>   $\pi = (s_1, \dots, s_k)$
>
>   $s_j = (\mathcal{T}_j, \text{args}_j)$
>
> **Q1. ​**Number of evaluation models.​​ ***(Lines 86-87, 107, 150-166, and 263 in Revision)***
>
> * We first clarify the **input and output** of Agent4Weakness, as described in lines 150-166. We first obtain a comprehensive evaluation dataset, $D_{eval}$, by benchmarking a large set of models on numerous datasets. $D_{eval}$ includes efficiency metrics (e.g., accuracy, inference time, token count), and instance-level data (questions, model responses, ground truth). Agent4Weakness accepts the query and $D_{eval}$ as the input and generates a report to answer the query.
> * In our experiments, we specifically **query the weaknesses of**​**8 target models** using the Q1-Q3 templates.
> * ​$D_{eval}$ **contains evaluation data for 104 models**​. This large pool is necessary to ensure that Agent4Weakness can perform a thorough and objective comparison against a wide range of relevant models (e.g., SOTA, or other models in the same series) when analyzing the weaknesses of a specific target model.
>
>
>
> **Q2.** How to ensure requests are representative and cover user needs?  ***​(Section 3 in Revision)***
>
> * Please see our response to ​**W2**​.

---

### Official Review · Reviewer_dRrW · 2025-11-01

**Soundness:** 2
**Presentation:** 3
**Contribution:** 3
**Rating:** 2
**Confidence:** 3

**Summary:**

This paper proposes AGENT4WEAKNESS, a multi-agent framework using three agents (Planner, Analyzer, Reporter) with 22 specialized tools to generate comprehensive evaluation reports identifying LLM weaknesses. The authors claim their approach addresses two key limitations in existing methods: insufficient statistical comparison and inflexible evaluation perspectives.

**Strengths:**

The work is generally interesting and provide novel perspective into LLM evaluation. The challenge of systematically discovering LLM weaknesses beyond simple accuracy scores is crucial for the field. LLM evaluation requires systematic yet novel benchmarking.

**Weaknesses:**

1. Circularity induced when using LLM to generate AGENT4WEAKNESS reports and evaluate their quality. This may create inherent bias where the evaluator model may favor outputs from its own framework over others.

2. The framework still relies on human curated benchmarks. Does the model to evaluate still need to run on many benchmarks or the evaluation could be made on partial results? Is inference computation saved by this system?

3. Can the system provide novel evaluations beyond available benchmarks? For example, if I would like to assess an agent's ability in creative writing, would the system still be applicable?

**Questions:**

1. Does the system exhibit consistent performance when analyzing the same model multiple times?

2. Running benchmarks and evaluations may incur significant computational overhead. Can this system run stably on research computational infrastructure?

---

> ### Author Response · Authors · 2025-11-17
> **Response to Reviewer dRrW**
>
> Thank you for recognizing our work. Below are our responses to your concerns. We have marked the revisions in **blue** in the manuscript.
>
> **W1.​** The evaluator model may favor outputs from its own framework over others.​​ ***(Appendix G.5 in Revision)***
>
> * We would like to demonstrate from two perspectives why this potential bias does not undermine the validity of our proposed method.
>
> 1. **Agent4Weakness focuses on identifying objective weaknesses. ​**Our primary goal is to leverage the agents to identify *objectively* verifiable weaknesses in models, rather than subjective preferences. We differentiate these two types of weaknesses:
>    1. **Objective Weaknesses:** These refer to quantifiable deficiencies, such as clear performance gaps (e.g., lower rankings or scores on specific benchmarks compared to peers) or distinct behavioral flaws (e.g., factual inaccuracies or severe hallucinations in responses).
>       * Our main experiments show that reports generated by Agent4Weakness achieve an improvement of 2.6 out of 10 across four dimensions compared with the baseline, with high consistency with human evaluations. Also, model performance improves by 3.7 when guided by the weakness discovered by Agent4Weakness, highlighting the potential for practical applications of our method. This confirms our method's ability to identify actionable, objective issues.
>    2. **Subjective Weaknesses:** These relate to preferences, such as disliking a model's linguistic style or its perceived unsuitability for a particular task.
>       * As these weaknesses lack standardized evaluation metrics [1], they are explicitly outside the scope of our current work.
> 2. **Data composition minimizes model-specific bias. ​**The input data for any single model constitutes an extremely small fraction of the entire dataset. Our input consolidates data from ​**104 models and 27 benchmarks**​. The input for each model is not limited to its generated responses, but also includes a wide array of ​**objective, model-agnostic statistics**​, such as token consumption, latency, and other computational metrics. These statistical features are inherently objective and do not carry any model-specific characteristics that an evaluator could be biased towards. This diverse and statistically-grounded input further migitate the potential for self-favoritism.
>
> **W2.** Our method's reliance on the full benchmark and whether it saves inference costs. ***(Lines 75-85, Appendix G.3 in Revision)***
>
> * Our method does not rely on the full benchmark and it is not designed to reduce inference costs. We explain these below.
> 1. Our framework identifies weaknesses ​**based on evaluation results**​. It does not require running the complete benchmark; users can selectively run evaluations on specific areas of concern and input those results into Agent4Weakness.
> 2. We would like to highlight the following motivation. Discovering weaknesses is crucial for model improvement. However, standard benchmark scores alone are insufficient for this purpose. Therefore, we propose Agent4Weakness to generate detailed, user-specific weakness reports based on these evaluation results. Therefore, **Agent4Weakness is not designed to reduce inference costs**.
>     * Using our main experiment setting, we compare the average cost for Q3 (which requires the most resources due to extensive analysis across multiple benchmarks) among the baseline, Agent4Weakness, and human evaluators.
>     * For human evaluation, we engaged professional LLM evaluators to annotate reports for Q3 across 8 models (with time roughly estimated). While human-written reports score higher on all four aspects, they incur significant time costs. Furthermore, ​**human analysis is often limited by individual perspectives and struggles to scale**​, frequently overlooking weaknesses in large-scale instance analysis, which results in a lower "Content Value" score. Conversely, the 'Direct QA' baseline is efficient but performs poorly across all four dimensions.
>     * Thus, Agent4Weakness is a significant step towards an automated, flexible, accurate, and high-quality process for weakness identification.
>
> | **Method** | **Time** | **Input Tokens** | **Output Tokens** | **Requirement Fulfillment** | **Content Value** | **Factuality** | **Readability** |
> | :--- | :---: | :---: | :---: | :---: | :---: | :---: | :---: |
> | Direct QA | 21.7s | 109,107.9 | 1,912.3 | 6.9 | 6.6 | 7.3 | 8.3 |
> | Agent4Weakness | 246.3s | 107,013,483.3 | 15,336.7 | 8.9 | 8.7 | 8.1 | 8.7 |
> | Human Annotator | 30h | - | - | 10.0 | 9.0 | 10.0 | 9.5 |
>
> [1] Large Language Models for Subjective Language Understanding: A Survey

---

> > ### Author Response · Authors · 2025-11-17
> > **Response to Reviewer dRrW**
> >
> > **W3.** Can the system provide novel evaluations beyond available benchmarks? ***(Lines 75-85)***
> >
> > * Our framework requires evaluation results as input. For instance, to evaluate a model's creative writing ability where no standard benchmark exists, a user **can generate multiple creative writing samples** from the model and input these into Agent4Weakness for assessment.
> > * In response to W2, our framework aims to identify in-depth model weaknesses ​**based on existing evaluation results**​, rather than to create new datasets. Therefore, developing novel evaluation frameworks beyond available benchmarks is outside the scope of this paper.
> >
> > **Q1.** Does the system exhibit consistent performance when analyzing the same model multiple times? ***(Appendix G.4 in Revision)***
> >
> > * Yes, Agent4Weakness **exhibits consistent performance** when analyzing the same model multiple times.
> > * To assess consistency, we run Agent4Weakness 5 times to evaluate the same model (GPT-5-high) on the same queries (Q1, Q2, Q3). To prevent potential biases from LLM-based scoring, we employ human evaluation for these results. We present the table below (RF: Requirement Fulfillment, CV: Content Value, F: Factuality, R: Readability).
> > * The **low variance** quantitatively confirms that Agent4Weakness produces stable and reliable results, minimizing random fluctuations across independent runs.
> >
> > | | **Q1-RF** | **Q1-CV** | **Q1-F** | **Q1-R** | **Q2-RF** | **Q2-CV** | **Q2-F** | **Q2-R** | **Q3-RF** | **Q3-CV** | **Q3-F** | **Q3-R** |
> > | :--- | :---: | :---: | :---: | :---: | :---: | :---: | :---: | :---: | :---: | :---: | :---: | :---: |
> > | run1 | 8.0 | 8.0 | 9.0 | 8.0 | 9.0 | 9.0 | 8.0 | 9.0 | 8.0 | 8.0 | 8.0 | 9.0 |
> > | run2 | 9.0 | 8.0 | 8.0 | 8.0 | 10.0 | 9.0 | 8.0 | 8.0 | 9.0 | 9.0 | 8.0 | 9.0 |
> > | run3 | 9.0 | 8.0 | 8.0 | 8.0 | 10.0 | 9.0 | 8.0 | 8.0 | 9.0 | 8.0 | 10.0 | 9.0 |
> > | run4 | 9.0 | 9.0 | 8.0 | 8.0 | 10.0 | 9.0 | 8.0 | 9.0 | 9.0 | 9.0 | 8.0 | 10.0 |
> > | run5 | 9.0 | 8.0 | 8.0 | 8.0 | 10.0 | 9.0 | 9.0 | 8.0 | 9.0 | 9.0 | 8.0 | 9.0 |
> > | **Avg** | 8.6 | 8.2 | 8.2 | 8.0 | 9.8 | 9.0 | 8.2 | 8.4 | 8.6 | 8.6 | 8.4 | 9.2 |
> > | **Std** | 0.50 | 0.45 | 0.45 | 0.00 | 0.45 | 0.00 | 0.45 | 0.55 | 0.50 | 0.55 | 0.89 | 0.45 |
> >
> > **Q2.** Can this system run stably on research computational infrastructure? ***(Appendix G.3 in Revision)***
> >
> > * Yes, the system can **run stably** on standard research infrastructure.
> > * Agent4Weakness runs by calling the Claude-Opus-4.1-thinking API and only requires a CPU. For details on its computational cost and stability, please see our responses to W2 and Q1, respectively.

---

> > > ### Comment · Reviewer_dRrW · 2025-11-27
> > > **Thank you for your clarifications**
> > >
> > > Thank you for the clarifications and the explanation of your motivation. The additional experiments address my concern about evaluation consistency, and I will increase my rating to 4.
> > >
> > > I have rechecked the revised draft and prompts in the Appendix. I believe this kind of system would be useful for long and difficult-to-understand outputs, such as behavior analysis of agent trajectories. However, its reliance on benchmark questions may limit its applicability, as it would be hard to discover new weaknesses without suitable benchmarks, which are difficult to curate. Therefore, I am still leaning toward rejection.

---

> > > > ### Author Response · Authors · 2025-11-28
> > > >
> > > > Thank you for recognizing the utility of our system and increasing your score. We truly appreciate your engagement. Regarding your concern that "reliance on existing benchmarks limits the discovery of new weaknesses," we would like to clarify our ​**research scope and motivation**​, which we believe directly address this issue.
> > > >
> > > > **1. Motivation: Beyond scalar scores to granular diagnosis**
> > > >
> > > > Modern models often achieve highly similar scores on leaderboards (e.g., <1% difference in accuracy), yet their utility in real-world applications varies drastically. This discrepancy exists because scalar metrics treat the benchmark as a "Black Box," masking the qualitative reasons behind failures.
> > > >
> > > > * **Our Scope:** We advocate shifting the community's focus from merely *ranking* models to deeply *understanding* specific weaknesses. Agent4Weakness is designed to bridge this gap by **mining "new" capability and behavioral insights from "old" data**.
> > > >
> > > > **2. The necessity of automation**
> > > >
> > > > You noted that benchmarks are hard to curate. We argue that analyzing existing benchmarks is even harder without automation, which necessitates our tool.
> > > >
> > > > * Consider complex benchmarks like **SWE-bench** or agentic tasks, where a single interaction trajectory can exceed ​**tens of thousands of tokens**​.
> > > > * It is challenging for humans to manually inspect these massive logs to find patterns. If we do not use tools like Agent4Weakness, these benchmarks remain opaque "black boxes." Our method opens this box, transforming overwhelming raw data into actionable insights.
> > > >
> > > > **3. Discovering new weaknesses via standard benchmarks**
> > > >
> > > > Considering that that benchmarks are hard to curate, we demonstrate that we do not need new questions to find new problems. In the main experiments, Agent4Weakness **autonomously identified latent behavioral flaws** in SOTA models that were ​**neither prompted for nor labeled in the dataset**​:
> > > >
> > > > * **Gemini-2.5-pro:** Exhibits a "Content-over-Format" bias, systematically ignoring formatting constraints in complex prompts, and hallucinates cross-associations between expert entities in niche domains.
> > > > * **Deepseek-V3.1-0821-Thinking:** Shows rigid cognitive resource allocation (poor meta-cognition), applying identical reasoning depth regardless of problem complexity.
> > > > * **Seed-1.6-Thinking:** Displays objective misalignment, prioritizing theoretical verification over functional code implementation.
> > > >
> > > > **4. Quantitative proof of new weakness discovery**
> > > >
> > > > To rigorously prove this capability of Agent4Weakness, we conduct an additional analysis (see Table below). We measure the precision of Agent4Weakness in identifying specific behavioral patterns that are, crucially, **neither mentioned in our system prompts nor labeled in the benchmark datasets**. They are autonomously discovered by Agent4Weakness as "new" weaknesses arising from the model's interaction with standard questions.
> > > >
> > > >
> > > > | Pattern Discovered     | Emoji Abuse | Code-Switching | Overuse of Code Blocks | Excessive Self-Reflection |
> > > > | ------------------------ | ------------- | ---------------- | ------------------------ | --------------------------- |
> > > > | Detection Accuracy (%) | 100         | 100            | 83.3                   | 86.7                      |
> > > >
> > > > In summary, our ultimate goal is to **advance model development**, not just evaluate it. By sharing this methodology, we hope to empower the community to stop treating benchmarks as black boxes and start fixing the fundamental, qualitative defects in models. We believe this contribution offers significant value to the field.

---

### Comment · Area_Chair_MA3V · 2025-11-28
**Reminder: Engage with Authors During Rebuttal**

Dear reviewers, please engage with the authors during the rebuttal if you haven’t yet. The deadline is approaching; add clarifications and follow-ups in the submission thread to ensure a fair, informed decision. Thank you

---

### Author Response · Authors · 2025-11-28
**Rebuttal Summary for AC**

Dear AC,

We thank the reviewers and the AC for their time and thoughtful efforts on this submission.To help reduce the workload, we briefly summarize below the main content of our paper, the key reviewer concerns, and our responses.

---

Our work introduces **Agent4Weakness**, a tool-augmented **multi-agent system for diagnosing LLM weaknesses**. Given a large evaluation corpus (104 models, 27 benchmarks, efficiency metrics and instance-level traces) and a user query, our Planner–Analyzer–Reporter agents cooperate to turn existing evaluations into *tructured, evidence-based weakness reports for a target model. These reports are not just descriptive: when used as in-context guidance, they yield an average **3.7-point performance gain** without any retraining. Agent4Weakness consistently **outperforms both Direct QA and a strong One-Agent+tools baseline** on Requirement Fulfillment, Content Value, Factuality, and Readability, across multiple backbone LLMs with high consistency with human evaluations.

---

Key concerns and our responses (post-rebuttal version of the paper reflects these):

1. **Novelty and contribution scope** (S6xB)
We clarify that our main contribution is a **paradigm shift from leaderboard scores to scalable, fine-grained weakness diagnosis** over large evaluation logs. Extensive ablations and a strong One-Agent baseline (with identical tools) show that our multi-agent design and role structure are crucial to the large quality gains, and that the framework directly closes the loop to **practical, in-context model improvement**. We advocate shifting the community's focus from merely *scoring* models to deeply *understanding* specific weaknesses.

2. **Reliance on existing benchmarks**  (dRrW)
Our explicit scope is to **mine new behavioral and capability insights from existing data**, not to design new benchmarks. We show that Agent4Weakness **autonomously uncovers previously unlabeled behavioral patterns** (e.g., emoji abuse, code-switching, overuse of code blocks, excessive self-reflection) with high detection accuracy. We also focus on **objective weaknesses** and mitigate evaluator bias by operating on a diverse corpus of 104 models with largely **model-agnostic statistics**.

3. **Query diversity and representativeness**  (S6xB)
We explain that Q1–Q3 correspond to the three dominant practitioner needs (performance, efficiency, behavior). A survey of **51 LLM practitioners** shows these three categories cover ~99.5% of real-world diagnostic requirements. We further add **three more specific queries (Q4–Q6)** and confirm that the gains persist, demonstrating robustness to a wider variety of weakness-oriented questions.

4. **Ablations, baselines, and model sensitivity**  (S6xB, 4zmv)
We add a One-Agent+tools baseline and extended ablations on roles (Planner/Analyzer/Reporter) and tool families. Removing roles or deep analysis tools significantly degrades performance, especially on complex behavioral queries. Running Agent4Weakness with Claude, GPT, and Gemini as backbones consistently yields strong improvements over Direct QA, indicating that the framework is robust to the underlying LLM.

5. **Complexity, cost, and practicality; nature of “improvement”**  (dRrW, DEcy, 4zmv)
We clarify that Agent4Weakness is **not designed to reduce inference cost**; it operates as a post-hoc analysis layer on (possibly partial) evaluations. Compared to ~30 hours of expert human analysis for Q3, our system is **far more scalable** while achieving comparable quality. It runs stably on **standard research infrastructure** via API (CPU-only on the user side). We explicitly state that all reported gains are in-context improvements, which we view as a strength because they are **immediately actionable for practitioners** without retraining.

We respectfully believe that, with these clarifications and additional experiments, our submission offers a **conceptually meaningful, empirically solid, and practically useful** step toward turning evaluation from “scores on a leaderboard” into **actionable, high-resolution model diagnostics**.

---

We again thank the AC for their careful handling of this paper and for their service to the community.

Best Regards,

Authors

---

### Note · Authors · 2026-01-05

I have read and agree with the venue's withdrawal policy on behalf of myself and my co-authors.